

# Personality Assessment Inventory (PAI): obsolete norms identify psychopathology in nearly everyone

Bob Uttl, Kiefer Sikma and Mikayla Tat

Department of Psychology, Mount Royal University, Calgary, AB, Canada

## ABSTRACT

**Background.** The Personality Assessment Inventory (PAI), a self-report personality test, is one of the frequently used measures to assess psychopathology in a wide variety of settings including in high stakes assessments, for example, in child custody disputes, employment settings and fitness for duty assessments. The PAI has never been normed on a Canadian population and the PAI users have simply assumed that the US norms also describe the Canadian population. Moreover, accumulated research demonstrates that the PAI's 35 years outdated and obsolete norms no longer describe neither university students' nor normal adult US populations.

**Method.** We administered the PAI to over 200 university students in a mid-size Canadian university.

**Results.** Our students scored on average in moderately elevated range (60T to 69T) on many of the PAI scales including anxiety (ANX), anxiety-related disorders (ARD), depression (DEP), schizophrenia (SCZ), and borderline features (BOR). Multivariate base rate analyses revealed that approximately 95% of our sample scored in elevated range on at least one out of the 22 PAI Scales. Furthermore, although some of the PAI reliabilities are adequate for research, the PAI reliabilities are too low for using the PAI in high stakes and forensic assessment, for example, in insurance benefits, child custody, employment, and fitness for duty assessments.

**Discussion.** We conclude that the PAI US norms are no longer appropriate for high-stakes assessments, ought to be withdrawn immediately, and new up-to-date norms ought to be established to prevent mislabelling and diagnostic misclassifications of and harm to examinees. Continued use of the PAI outdated norms in high stakes assessments carries ethical risks, is non-scientific, and likely amounts to malpractice.

Corresponding author
Bob Uttl, uttlbob@gmail.com

## INTRODUCTION

The Personality Assessment Inventory (PAI) (*Morey, 1991*; *Morey, 2007*), a self-report 344-item personality test, is one of the frequently used measures to assess psychopathology. The PAI Scales include four validity scales (ICN = Inconsistency, INF = Infrequency, NIM = Negative Impression Management, PIM = Positive Impression Management), 11 clinical scales (SOM = Somatic Complaints, ANX = Anxiety, ARD = Anxiety-Related

Disorders, DEP = Depression, MAN = Mania, PAR = Paranoia, SCZ = Schizophrenia, BOR = Borderline Features, ANT = Antisocial Features, ALC = Alcohol Problems, DRG = Drug Problems), five treatment scales (AGG = Aggression, SUI = Suicidal Ideation, STR = Stress, NON = Nonsupport, RXR = Treatment Rejection), two interpersonal scales (DOM = Dominance, WRM = Warmth), and numerous additional supplementary scales and indexes.

In this paper, we first examine published research on the PAI demonstrating that (a) the PAI's original US norms are outdated and no longer describe the US population, (b) the PAI's US norms did not generalize to other countries shortly after their publication in 1991, (c) the PAI's scales, subscales, and various supplementary indexes are not sufficiently reliable to be used in high-stakes decisions about individuals, and (d) there are no published data on multivariate base rates of elevated and high scores, that is, $T$-score $\geq 60T$ and $\geq 70T$, on the PAI scales. Second, we report on a new study that examined (a) the performance of the Canadian university students on the PAI, (b) reliabilities of the PAI scales and subscales, and (c) univariate and multivariate base rates of elevated and high scores. Finally, we discuss the implications of our findings including obsolescence of the PAI, insufficient reliabilities of the PAI scales and subscales, and very high multivariate base rates of elevated and high scores when using the US outdated and obsolete norms as well as the current performance of our Canadian university students.

The PAI manual provides norms based on three US samples: a census matched adults sample, a clinical sample, and a college students sample. Notably, the US college sample did not differ substantially from the subsample of 20–29 year old US adults in 1990. All three US standardization samples were tested prior to 1991, approximately 35 years ago, and accordingly, the PAI US norms are now about 35 years outdated. Several large sample studies with US college students indicate that the US college students norms are now obsolete and no longer describe today's US college students. *Nails et al. (2023)* administered the PAI to large samples of US university students and found that the students' scores on the PAI have shifted towards higher degree of psychopathology prior to the COVID-19 pandemic ($N = 558$; tested between September 2016 and November 2019 at Villanova University, University of South Dakota, and Mississippi State University) and that they shifted still further after the COVID-19 pandemic ($N = 825$; tested between September 2021 and April 2022 at Texas A&M University and Villanova University) (*Nails et al., 2023*) (note that the US university students tested with the PAI in another large sample study (*Maffly-Kipp & Morey, 2024*) were included in *Nails et al. (2023)* post COVID-19 sample). The shifts reported by *Nails et al.* are non-trivial, approximately one standard deviation (SD) or 10T (T = $T$-scores standardized with $M = 50$ and $SD = 10$) equivalent, with the largest increases observed on the ANX, ARD, and DEP scales. Accordingly, *Nails et al.* samples (with a combined sample larger than the original US normative college student sample) strongly indicate that the US 1990 norms, at least for the US university students, are now outdated, obsolete and no longer appropriate for use with the university students in the US. Similarly, the most recent data collected online from US adults indicate that even the US norms for the US census matched adult samples are antiquated and no longer appropriate, with the adults scoring substantially higher on a number of the PAI scales

including ANX (4.8T higher), ARD (4.8T higher), and DEP (7.6T higher) (*McCredie & Morey, 2019*).

Moreover, a number of studies reported that when the PAI or PAI translations were used in other countries, the mean scores of these foreign samples differed substantially from the US norms (*Aschieri et al., 2024*; *Boyle & Lennon, 1994*; *Groves & Engel, 2007*; *Jeffay et al., 2021*; *Paulino et al., 2024a*) indicating that the US norms (*Morey, 1991*; *Morey, 2007*) did not describe samples in these countries at the time the samples were tested, either because the US norms were never applicable to such countries, or because the US norms became outdated and obsolete, or both. For example, *Boyle & Lennon (1994)* reported that their sample of 170 college students and community dwelling adults in Australia scored much higher than the US normative samples on many of the scales including ALC (63T), STR (61T), and SUI (60T). *Aschieri et al. (2024)* reported that a nonclinical sample of 58 Italian adults scored up to 10 *T* scores equivalent above the US norms on the Italian translation of the PAI. In Canada, *Jeffay et al. (2021)* reported that 50 English/French bilingual university students scored up to 5T scores above the US norms on some of the clinical scales on both the original English version as well as the French adaptation of the PAI, and that the students' scores on English *vs.* French adaptations also differed substantially. Most recently, *Paulino et al. (2024a)* reported that a Portuguese normative sample (with the normative data collected online) scored substantially higher than the US norms on numerous scales including ANX, DEP, PAR, and BOR.

Figure 1 highlights that the PAI US norms do not necessarily generalize to other countries and cultures; it shows the difference in the PAI performance of adult samples collected in different countries/cultures relative to the US 1990 adult norms using Cohen's d. Whereas in some countries such as Greece the performance of the Greek standardization sample was mostly similar to the US norms, in other countries the performance was substantially different from the US norms. Some of the differences may be due to translation of the PAI but, as noted above, the Australian sample tested shortly after the publication of the PAI scored substantially higher on many of the PAI scales on the original English version of the PAI. A large variability in the PAI scores across countries/cultures ought not to come as a surprise. For example, a large study investigating geographical distribution of the big five personality traits in 56 nations revealed that performance on the Big Five Inventory varied substantially across countries and cultures (*Schmitt et al., 2007*). More recently, *Dong & Dumas (2020)* investigated the measurement invariance across cultures using meta-analysis of previous studies and concluded that none of the personality measures investigated showed measurement invariance across cultures. In turn, these findings make it clear that personality scores obtained by examinees in one country or culture ought not to be interpreted using norms established in different country or culture. Figure 1 also shows the performance of the US adult sample tested just prior to COVID-19 pandemic scored much higher than the US adult normative sample with the mean scores differences exceeding 0.5 SD on ANX, ARD, DEP, PAR, SUI, and WRM scales. Collectively, these findings strongly suggest that the US norms are likely (a) outdated and obsolete for both the US normal adults and university students, (b) not appropriate for different populations, different countries/cultures, and (c) not appropriate with translated versions of the PAI.

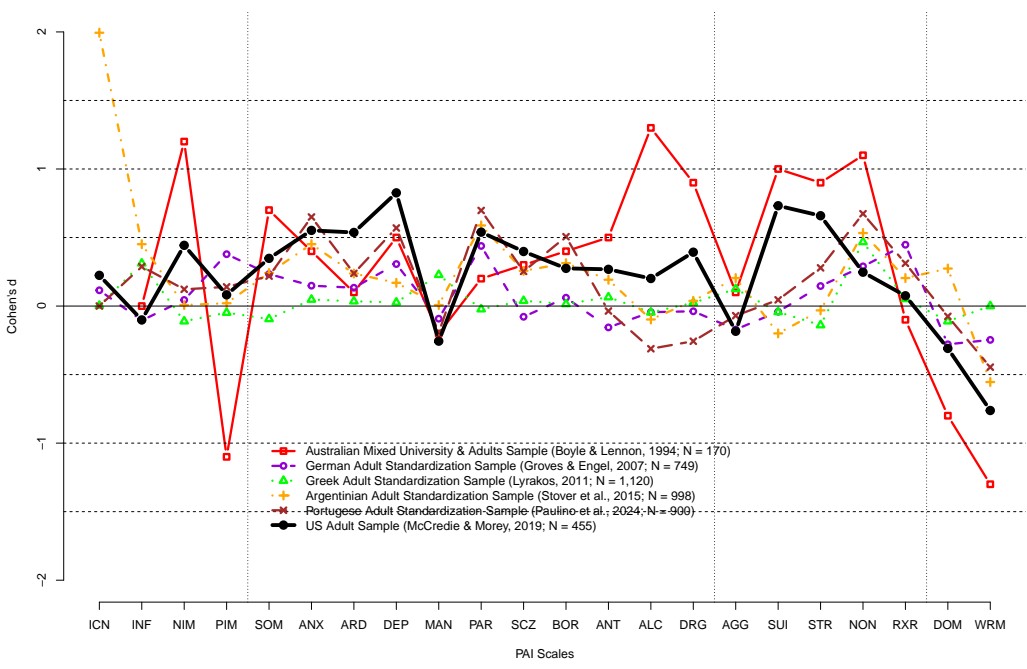

**Figure 1** **The PAI scale profile shifts of adult community samples collected in different countries at different times in terms of Cohen's d.** The difference between the US 1990 adult normative sample and samples collected in different countries and different times. Australian sample demonstrates substantially different profile shortly after the PAI publication. In contrast, Greek adult profile was relatively similar to the PAI 1990 adult profile. Notably, the US adult sample profile collected prior to the COVID-19 pandemic revealed large shifts ($d > 0.50$) on a number of scales including ANX, ARD, DEP, PAR, SUI, STR, and WRM scales.

Although *Boyle & Lennon*'s (*1994*) results showed that the US norms did not even generalize to an English speaking Australian population shortly after the PAI was published, the PAI has never been normed on Canadian populations of adults, clinical samples, or college students samples. Canadian psychologists have simply assumed that the Canadian norms would be equivalent and thus, appropriate for use in Canada, and have been using the PAI in both low stakes and high-stakes decisions such as screening applicants for jobs, determining insurance benefits, fitness-for-duty evaluations, and criminal proceedings based on the US norms. For example, searching the Canadian legal cases depository canlii.org, for "Personality Assessment Inventory" revealed 977 cases in which the PAI was referred to and used in published decisions. Although psychometric theory makes it clear that clinical psychologists ought not to use norms developed in different countries as those norms are not necessarily applicable in other countries, a survey of Canadian neuropsychologists revealed that eighty percent of them use norms from other countries to score neuropsychological tests (*Monette et al., 2023*).

However, there are several reasons why the US norms are most likely inappropriate for use in Canada today. First, the US studies reviewed above demonstrate that at least the US college student norms no longer describe today's US college students, are obsolete, and ought not to be used even in the US. Second, the studies with PAI in different countries,

including Australia, demonstrated that the US norms were a poor characterization of other countries' non-clinical samples, including Australia's, shortly after they were published. In turn, this suggests that the use of the US norms was likely also inappropriate in Canada. Third, the US large sample studies with both the university students and community adults strongly suggests that the US norms have changed in part due to the COVID-19 pandemic. Moreover, as detailed above, similar results have been reported in other countries. Fourth, the characteristics of the population of college students have changed over the last 35 years due to a growing proportion of the population pursuing university education and degrees (*Uttl, Violo & Gibson, 2024*) with most of the young adults pursuing university education and degrees today. Fifth, numerous related sources have reported increases in anxiety, depression and other mental health issues in adults, young adults, and university students. For example, Statistics Canada reported that the 12-month prevalence of depression, anxiety, and anxiety related disorders increased substantially between 2012 and 2022: nearly doubled for Major Depressive Episode (4.7 *vs.* 7.6%), doubled for Generalized Anxiety Disorder (2.6 to 5.2%), and more than doubled for Social Phobia (3.0 to 7.1%) (*Stephenson, 2023*). The analyses of the National College Health Assessment II survey data revealed large increases in the proportion of post-secondary students who self-reported symptoms of hopelessness, loneliness, sadness, depression, anxiety, anger, suicide considerations and other symptoms as well as diagnoses of anxiety, depression, sleep disorders and other disorders (*Linden, Boyes & Stuart, 2021*). Similarly, there have been massive documented increases in numbers of university students seeking accommodations for perceived impairments and disabilities (*Suhr & Johnson, 2022*).

In combination, the previous research strongly suggests that the PAI's 35 year outdated US norms do not describe Canadian populations today. In turn, the continued use of the PAI's outdated US norms is likely to result in misclassification of normal scores as abnormal scores indicating psychopathology, and likely to contribute to misdiagnoses not only in the US but also in Canada. Moreover, when the PAI is used in high-stakes assessments, for example, in forensic, employment and custody dispute situations, psychologists' interpretations of the "abnormal" PAI scores will likely be wrong and cause serious, and sometimes irreparable harm to the examinees and society.

It is also unclear whether reliabilities of the PAI scales and subscales would be comparable in Canadian *vs.* US normative samples. Independent research that examined reliabilities of the PAI scales found that the reliabilities vary widely, ranging from low .08 and rarely exceeding .90 (*Boyle & Lennon, 1994*; *Groves & Engel, 2007*; *Jeffay et al., 2021*; *Paulino et al., 2024a*). We were also unable to find any study that examined multivariate base rates of examinees in the normative samples who obtained one or more elevated PAI scale scores, that is, T-score ≥60T and ≥70T, considered "elevated" and "high" (*Morey, 2007*), respectively. It is well known from research on the interpretation of cognitive ability and intelligence test scores that more tests one administers and more scores one computes, the more "abnormal" scores one will observe among those scores *purely due to chance*, that is, due to random variation in observed scores due to their imperfect reliabilities (*Crawford, Garthwaite & Gault, 2007*; *Karr et al., 2017*; *Karr et al., 2018*; *Karr et al., 2020*). In turn, assessors who are unaware of multivariate base rates and who use the PAI with its

22 scales, numerous subscales, and numerous other supplementary indexes, are likely to misinterpret elevated and high scores caused by random variation in observed scores (due to scores' unreliability) as signs of personality psychopathology. In turn, these random error interpreters may unjustly malign examinees' reputations, and may even cause examinees to lose their job, or to lose custody of their children.

The main objectives of our study were to examine performance of Canadian university students on the PAI, to determine reliabilities of the PAI scales and indexes, to determine base rates of elevated and high scores for the PAI scales and subscales relative to our sample *vs.* the US norms, and to determine multivariate base rates of elevated and high scores relative to our sample *vs.* the US norms. To achieve these objectives, we administered the PAI to a large sample of Canadian university students.

## MATERIALS & METHODS

### Participants

The participants were 207 undergraduate students enrolled in introductory psychology classes in a mid-size university in Canada who participated in the study for a course credit. Students were administered the PAI following the PAI Manual instructions (paper & pencil administration) as part of a 2-hour long study on individual differences in cognition and personality. Participants were tested in small groups ranging from 1 to 10 students. The study was approved by the Mount Royal University Human Research Ethics Board (HREB #103697). Written informed consent was obtained from all participants.

Participants' data were excluded if they (a) left more than 17 items missing or (b) produced invalid profiles defined as ICN $\geq$ 13 (ICN $\geq$ 73T) or INF $\geq$ 9 (INF $\geq$ 75T) (*Morey, 2007*). In total, one participant was excluded due to having more than 17 missing item responses and two participants were excluded due to invalid profiles. Following these exclusions, 204 participants were left in the analysis (160 women, 38 men, one transgender female, zero transgender male, two gender variant, two prefer not to answer, 1 other; age in years: $M = 21.2$ years, $SD = 4.96$; race/ethnicity: 49.0% Caucasian, 13.7% Asian, 7.8% African, 7.4% Indian/Brown, 4.9% Hispanic/Latino, 5.9% Middle Eastern, 4.9% Native American, 5.4 Other, 0.9% Prefer not to say).

### Measures

The Personality Assessment Inventory (*Morey, 1991*; *Morey, 2007*) is a 344-item self-report measure of personality. It includes 4 validity scales, 11 clinical scales, 5 treatment scales, 2 interpersonal scales, numerous subscales and numerous other indexes. All items are answered on a 4-point scale: 0 = *False, not at all true*, 1 = *Slightly true*, 2 = *Mainly true*, 3 = *Very true*. Tables 1 and 2 shows the scales and subscales and their abbreviations. The PAI scales, subscales, and supplementary scales are calculated as described in the PAI manual, and the resulting $T$ scores ($M = 50$, $SD = 10$) reflect a comparison to the US 1990 adult community norms (*Morey, 2007*; p. 2). The authors have permission to use this instrument from the copyright holders (*Morey, 2007*).

Utl et al. (2025), *PeerJ*, DOI 10.7717/peerj.20340

**Table 1  Descriptive statistics for the US 1990 (*Morey, 1991*; *Morey, 2007*) and Canadian 2024/2025 university student sample for PAI Scales.**

| | US | | | | CDN | | | | | | | | | |
|---|---|---|---|---|---|---|---|---|---|---|---|---|---|---|
| | M | SD | $\alpha$ | $r_{xx}$ | M | SD | $\omega$ | $\alpha$ | SEm | d | $M_T$ | $SD_T$ | ≥60T % | ≥70T % |
| **Validity** | | | | | | | | | | | | | | |
| ICN | 4.92 | 2.69 | 0.26 | 0.32 | 5.80 | 2.41 | 0.08 | −0.04 | – | 0.33 | 51.40 | 7.22 | 15.20 | 0.98 |
| INF | 2.84 | 2.04 | 0.22 | 0.55 | 2.91 | 1.99 | 0.02 | 0.08 | 9.60 | 0.03 | 50.76 | 7.79 | 11.27 | 1.96 |
| NIM | 1.50 | 2.20 | 0.63 | 0.80 | 4.21 | 3.86 | 0.72 | 0.69 | 5.54 | 1.23 | 59.33 | 14.30 | 35.78 | 25.00 |
| PIM | 13.08 | 4.36 | 0.73 | 0.75 | 10.69 | 4.85 | 0.77 | 0.77 | 4.82 | −0.55 | 39.90 | 11.15 | 1.47 | 0.49 |
| **Clinical** | | | | | | | | | | | | | | |
| SOM | 7.86 | 6.92 | 0.83 | 0.81 | 19.95 | 13.17 | 0.90 | 0.90 | 3.11 | 1.75 | 58.95 | 13.17 | 40.69 | 22.06 |
| ANX | 18.46 | 10.32 | 0.89 | 0.88 | 34.25 | 15.17 | 0.93 | 0.93 | 2.71 | 1.53 | 66.81 | 14.36 | 63.73 | 42.16 |
| ARD | 19.20 | 8.78 | 0.80 | 0.84 | 33.22 | 11.15 | 0.82 | 0.83 | 4.12 | 1.60 | 66.05 | 13.39 | 66.18 | 36.76 |
| DEP | 13.20 | 8.68 | 0.87 | 0.86 | 27.12 | 13.98 | 0.92 | 0.92 | 2.90 | 1.60 | 63.60 | 14.87 | 55.88 | 30.88 |
| MAN | 27.21 | 9.48 | 0.82 | 0.76 | 29.37 | 10.06 | 0.81 | 0.81 | 4.34 | 0.23 | 56.88 | 10.94 | 37.25 | 14.22 |
| PAR | 18.87 | 8.57 | 0.86 | 0.83 | 25.93 | 10.51 | 0.86 | 0.86 | 3.74 | 0.82 | 58.58 | 12.13 | 45.10 | 17.65 |
| SCZ | 13.44 | 7.68 | 0.82 | 0.79 | 22.59 | 10.19 | 0.85 | 0.85 | 3.93 | 1.19 | 61.03 | 13.08 | 51.96 | 25.00 |
| BOR | 22.93 | 10.33 | 0.86 | 0.82 | 32.02 | 12.44 | 0.88 | 0.88 | 3.44 | 0.88 | 64.02 | 12.44 | 61.27 | 35.78 |
| ANT | 18.92 | 10.24 | 0.86 | 0.87 | 18.71 | 10.19 | 0.84 | 0.83 | 4.08 | −0.02 | 56.08 | 11.24 | 32.35 | 13.24 |
| ALC | 5.96 | 5.53 | 0.83 | 0.90 | 4.13 | 5.41 | 0.84 | 0.83 | 4.14 | −0.33 | 48.64 | 9.71 | 10.29 | 4.41 |
| DRG | 3.01 | 3.88 | 0.66 | 0.66 | 4.42 | 5.54 | 0.79 | 0.80 | 4.51 | 0.36 | 50.84 | 11.08 | 18.63 | 7.35 |
| **Treatment** | | | | | | | | | | | | | | |
| AGG | 16.48 | 9.69 | 0.89 | 0.78 | 14.72 | 10.04 | 0.89 | 0.89 | 3.34 | −0.18 | 49.90 | 11.90 | 21.08 | 9.80 |
| SUI | 3.92 | 5.20 | 0.87 | 0.85 | 9.05 | 8.50 | 0.93 | 0.92 | 2.80 | 0.99 | 61.79 | 17.59 | 47.06 | 27.45 |
| STR | 6.12 | 4.08 | 0.69 | 0.72 | 8.97 | 4.74 | 0.68 | 0.92 | 2.80 | 0.70 | 57.07 | 10.66 | 35.29 | 14.71 |
| NON | 4.43 | 3.56 | 0.75 | 0.74 | 6.96 | 4.58 | 0.77 | 0.78 | 4.64 | 0.71 | 55.62 | 12.48 | 31.86 | 14.71 |
| RXR | 14.12 | 4.16 | 0.72 | 0.73 | 10.35 | 4.56 | 0.75 | 0.73 | 5.16 | −0.91 | 42.68 | 9.83 | 4.90 | 0.00 |
| **Interpersonal** | | | | | | | | | | | | | | |
| DOM | 21.66 | 5.79 | 0.81 | 0.60 | 18.17 | 6.48 | 0.83 | 0.83 | 4.10 | −0.60 | 45.69 | 11.63 | 11.27 | 1.96 |
| WRM | 25.00 | 5.54 | 0.80 | 0.79 | 22.00 | 5.75 | 0.79 | 0.79 | 4.58 | −0.54 | 47.33 | 10.20 | 13.24 | 1.47 |

**Notes.**

ICN, Inconsistency; INF, Infrequency; NIM, Negative Impression Management; PIM, Positive Impression Management; SOM, Somatic Complaints; ANX, Anxiety; ARD, Anxiety-Related Disorders; DEP, Depression; MAN, Mania; PAR, Paranoia; SCZ, Schizophrenia; BOR, Borderline Features; ANT, Antisocial Features; ALC, Alcohol Problems; DRG, Drug Problems; AGG, Aggression; SUI, Suicidal Ideation; STR, Stress; NON, Nonsupport; RXR, Treatment Rejection; DOM, Dominance; WRM, Warmth; $\omega$, MacDonald's $\omega$; $\alpha$, Cronbach's $\alpha$; $r_{xx}$, test–retest reliability; SEm, Standard Error of Measurement; d, Cohen's d with SD in denominator based on the US 1990 norms; $M_T$, mean T-score based on US norms; $SD_T$, SD of the mean scores based on US norms; ≥60T % and ≥70T %, the percentage of scores equal or exceeding 60T and 70T, respectively.

US 1990 Norms are from *Morey (2007)* Table 4.13, Table 8.1, Table 8.7.

**Table 2** Descriptive statistics for US 1990 (*Morey, 1991*; *Morey, 2007*) and Canadian 2024/2025 university student samples for PAI Subscales (raw scores) and for Canadian 2024/2025 university student sample *T*-scores.

| | US | | | | | CDN | | | | | | | |
|---|---|---|---|---|---|---|---|---|---|---|---|---|---|
| | *M* | *SD* | *α* | $r_{xx}$ | *SEm (rxx)* | *M* | *SD* | *ω* | *α* | *SEm (α)* | *d* | $M_T$ | $SD_T$ |
| SOM-C | 1.55 | 2.29 | 0.60 | 0.66 | 5.7 | 5.50 | 5.20 | 0.81 | 0.81 | 4.41 | 1.72 | 58.87 | 15.25 |
| SOM-S | 3.48 | 3.09 | 0.64 | 0.70 | 4.6 | 7.75 | 4.53 | 0.72 | 0.71 | 5.37 | 1.38 | 58.63 | 12.09 |
| SOM-H | 2.84 | 3.02 | 0.70 | 0.84 | 4.4 | 6.70 | 5.63 | 0.87 | 0.87 | 3.62 | 1.28 | 56.07 | 13.27 |
| ANX-C | 7.12 | 4.61 | 0.83 | 0.85 | 3.9 | 12.50 | 5.89 | 0.86 | 0.85 | 3.81 | 1.17 | 64.96 | 13.60 |
| ANX-A | 6.87 | 3.86 | 0.73 | 0.80 | 4.6 | 11.65 | 5.11 | 0.79 | 0.77 | 4.76 | 1.24 | 64.11 | 13.33 |
| ANX-P | 4.48 | 3.18 | 0.66 | 0.83 | 4.1 | 10.10 | 5.53 | 0.83 | 0.82 | 4.22 | 1.77 | 66.75 | 15.58 |
| ARD-O | 8.52 | 4.07 | 0.67 | 0.74 | 5.1 | 12.76 | 4.18 | 0.64 | 0.61 | 6.24 | 1.06 | 59.02 | 11.18 |
| ARD-P | 6.42 | 3.40 | 0.58 | 0.66 | 5.6 | 9.69 | 3.91 | 0.60 | 0.57 | 6.57 | 0.96 | 58.25 | 10.80 |
| ARD-T | 4.26 | 4.46 | 0.85 | 0.79 | 4.2 | 10.76 | 6.55 | 0.89 | 0.89 | 3.32 | 1.46 | 66.73 | 15.94 |
| DEP-C | 4.50 | 3.50 | 0.78 | 0.78 | 4.8 | 8.66 | 5.47 | 0.87 | 0.86 | 3.71 | 1.19 | 62.63 | 16.00 |
| DEP-A | 3.91 | 3.35 | 0.79 | 0.83 | 3.9 | 8.60 | 5.59 | 0.87 | 0.87 | 3.59 | 1.40 | 62.44 | 15.23 |
| DEP-P | 4.78 | 3.39 | 0.64 | 0.74 | 5.0 | 9.86 | 4.74 | 0.67 | 0.68 | 5.65 | 1.50 | 59.49 | 11.40 |
| MAN-A | 7.75 | 3.46 | 0.57 | 0.54 | 5.7 | 9.66 | 3.70 | 0.50 | 0.52 | 6.91 | 0.55 | 59.25 | 11.60 |
| MAN-G | 10.01 | 4.73 | 0.79 | 0.77 | 4.4 | 9.04 | 4.92 | 0.78 | 0.78 | 4.68 | −0.20 | 51.54 | 11.31 |
| MAN-I | 9.45 | 4.35 | 0.78 | 0.79 | 4.6 | 10.67 | 4.98 | 0.81 | 0.80 | 4.44 | 0.28 | 56.52 | 11.71 |
| PAR-H | 8.07 | 3.67 | 0.72 | 0.77 | 4.7 | 10.85 | 4.33 | 0.74 | 0.74 | 5.12 | 0.76 | 59.47 | 12.76 |
| PAR-P | 3.51 | 3.21 | 0.77 | 0.68 | 5.0 | 5.48 | 3.98 | 0.72 | 0.71 | 5.29 | 0.61 | 55.38 | 11.78 |
| PAR-R | 7.30 | 3.32 | 0.65 | 0.75 | 5.1 | 9.60 | 4.02 | 0.66 | 0.65 | 5.87 | 0.69 | 56.81 | 11.40 |
| SCZ-P | 4.68 | 3.09 | 0.61 | 0.74 | 5.1 | 5.43 | 3.43 | 0.57 | 0.57 | 6.57 | 0.24 | 54.45 | 11.41 |
| SCZ-S | 4.14 | 3.56 | 0.80 | 0.83 | 4.1 | 7.62 | 4.81 | 0.82 | 0.82 | 4.28 | 0.98 | 55.18 | 12.24 |
| SCZ-T | 4.61 | 3.74 | 0.79 | 0.75 | 4.7 | 9.54 | 5.04 | 0.83 | 0.82 | 4.29 | 1.32 | 65.36 | 14.75 |
| BOR-A | 5.74 | 3.70 | 0.78 | 0.85 | 4.2 | 7.90 | 4.01 | 0.78 | 0.77 | 4.79 | 0.58 | 59.73 | 12.08 |
| BOR-I | 7.13 | 3.46 | 0.65 | 0.70 | 4.6 | 9.94 | 3.86 | 0.67 | 0.65 | 5.92 | 0.81 | 64.83 | 11.57 |
| BOR-N | 5.95 | 3.32 | 0.67 | 0.67 | 5.3 | 9.20 | 4.16 | 0.74 | 0.73 | 5.23 | 0.98 | 62.76 | 13.08 |
| BOR-S | 4.11 | 2.73 | 0.66 | 0.78 | 4.7 | 4.98 | 3.41 | 0.72 | 0.73 | 5.24 | 0.32 | 56.52 | 13.19 |
| ANT-A | 6.34 | 4.71 | 0.76 | 0.80 | 4.1 | 6.06 | 4.88 | 0.75 | 0.74 | 5.06 | −0.06 | 52.43 | 11.05 |
| ANT-E | 4.97 | 3.16 | 0.63 | 0.70 | 5.0 | 5.23 | 2.93 | 0.44 | 0.45 | 7.46 | 0.08 | 56.07 | 9.72 |
| ANT-S | 7.61 | 4.35 | 0.77 | 0.84 | 3.9 | 7.42 | 4.62 | 0.75 | 0.75 | 5.03 | −0.04 | 57.30 | 12.57 |
| AGG-A | 5.86 | 4.01 | 0.80 | 0.68 | 5.3 | 5.32 | 4.59 | 0.87 | 0.86 | 3.73 | −0.13 | 48.71 | 12.75 |
| AGG-V | 7.97 | 4.01 | 0.77 | 0.74 | 5.0 | 6.59 | 4.03 | 0.75 | 0.75 | 4.96 | −0.34 | 49.60 | 11.51 |
| AGG-P | 2.65 | 3.33 | 0.79 | 0.86 | 4.8 | 2.81 | 3.31 | 0.76 | 0.74 | 5.10 | 0.05 | 51.72 | 11.25 |

**Notes.**

SOM, Somatic Complaints; SOM-C, Conversion; SOM-S, Somatization; SOM-H, Health Concerns; ANX, Anxiety; ANX-C, Cognitive; ANX-A, Affective; ANX-P, Physiological; ARD, Anxiety-Related Disorders; ARD-O, Obsessive-Compulsive; ARD-P, Phobias; ARD-T, Traumatic Stress; DEP, Depression; DEP-C, Cognitive; DEP-A, Affective; DEP-P, Physiological; MAN, Mania; MAN-A, Activity Level; MAN-G, Grandiosity; MAN-I, Irritability; PAR, Paranoia; PAR-H, Hypervigilance; PAR-P, Persecution; PAR-R, Resentment; SCZ, Schizophrenia; SCZ-P, Psychotic Experiences; SCZ-S, Social Detachment; SCZ-T, Thought Disorder; BOR, Borderline Features; BOR-A, Affective Instability; BOR-I, Identity Problems; BOR-N, Negative Relationships; BOR-S, Self-Harm; ANT, Antisocial Features; ANT-A, Antisocial Behaviors; ANT-E, Egocentricity; ANT-S, Stimulus Seeking; AGG, Aggression; AGG-A, Aggressive Attitude; AGG-V, Verbal Aggression; AGG-P, Physical Aggression; ω, MacDonald's ω; α, Cronbach's α; $r_{xx}$, test–retest reliability; SEm ($r_{xx}$), Standard Error of Measurement based on $r_{xx}$; SEm (α), Standard Error of Measurement based on α; d, Cohen's d with SD in denominator based on the US 1990 norms.

US Norms are from *Morey (2007)* Table 4.14, Table 8.2, Table 8.8.

## Statistical analyses

All statistical analyses were performed using R (*R Core Team, 2021*). We examined reliability of scales, subscales and supplementary scales using Cronbach's alpha to ensure comparability of our findings with those reported in the PAI Manual (*Morey, 2007*) and other published research reviewed in the introduction. However, we also provide McDonald's omega designed to address some of the limitations of Cronbach's alpha. Importantly, we note that McDonald's omega has problems of its own (*e.g.*, estimates fail to compute), and, more importantly, the simulation studies show that McDonald's omega and Cronbach's alpha result in nearly identical estimates when sample sizes are reasonably large, reliabilities moderate to high, and scales have more than a few items (*Edwards, Joyner & Schatschneider, 2021*).

## RESULTS

Figure 2 shows the PAI scale profile of our Canadian university students sample. Each boxplot shows the distribution of students' scores for a specific PAI scale. The box plots show the median (thick line), 25th and 75th percentile (box edges), extent of non-outlying scores (whiskers), and outliers (*i.e.,* values defined as more than 1.5 inter-quartile range from 25th and 75th percentiles). The dashed red line shows the mean of the US normative sample tested in 1990, that is, the *T*-score of 50. Our data show that Canadian university students' profile today is marked by a very high degree of psychopathology relative to the US general population norms with the median scores of many PAI scales including anxiety (ANX), anxiety-related disorders (ARD), depression (DEP), schizophrenia (SCZ), borderline features (BOR), falling in moderately elevated range (the median T score of 60T to 69T).

Figure 3 shows the PAI scale profile of the mean T scores for our Canadian university students relative to the profile of Canadian bilingual university students reported by *Jeffay et al. (2021)*, the 1990 profile of the US college students reported in the PAI manual (*Morey, 2007*), and the profiles of the US college students reported more recently by *Nails et al. (2023)*. Our Canadian university students' profile indicates high elevation on many of the scales relative to both the PAI 1990 US college students norms as well as relative to the recent US college student profiles reported by *Nails et al. (2023)*.

Figure 4 shows the PAI scale profiles using Cohen's d, including 95% confidence intervals for the d, relative to the US 1990 university students norms (rather than the US 1990 adult norms). The figures reveal shifts towards higher scores on the majority of the PAI scales (*e.g.*, SOM, ANX, ARD, DEP, SCZ, SUI, BOR, PAR, STR, NON), several shifts towards lower scores (*e.g.*, PIM, ALC, RXR, DOM, WRM), and several scales with no or only relatively small shifts (*e.g.*, ICN, INF, MAN, ANT, AGG). Figure 5 shows the comparable data for the PAI subsales.

Table 1 shows the PAI Scales descriptive statistics for the US college students norms and the current Canadian university sample including raw means, standard deviations, McDonald's omegas, Cronbach's alphas, and the effect size *d* for the mean difference between the US college students norms and the current Canadian university sample. The
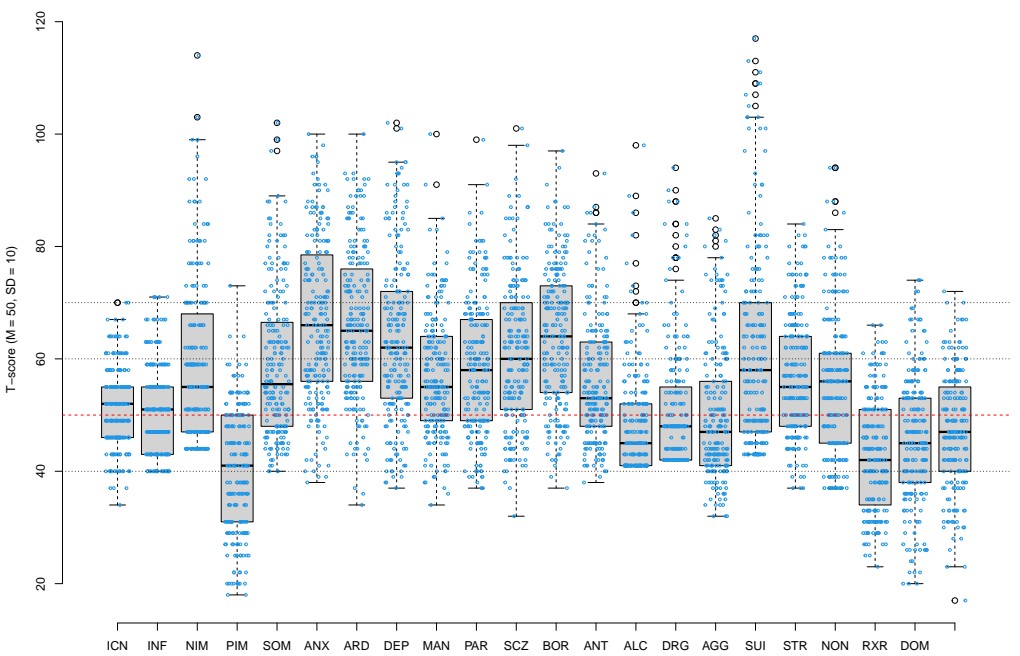

**Figure 2** **The PAI Scale profile of our Canadian university students sample.** Each boxplot shows the distribution of students' scores for a specific PAI scale. The box plots show the median (thick line), 25th and 75th percentile (box edges), extent of non-outlying scores (whiskers), and outliers (*i.e.,* values defined as more than 1.5 inter-quartile range from 25th and 75th percentiles). The dashed red line shows the mean of the US normative sample tested in 1990, that is, the *T*-score of 50. Overlayed over each box plot are individual data points jittered along horizontal axes only.

table also shows the current Canadian university sample *T*-score means and SDs, and percentage of students with *T* scores ≥ 60, that is with at least moderately elevated scores, and with *T* scores ≥ 70, that is high scores indicating significant concerns relative to the US norms. Moreover, multivariate base rates computation showed that 95% of Canadian students scored ≥ 60T on at least one of the 22 PAI scales and 73% scored ≥ 70T on at least one of the 22 PAI scales. Only 10 out of 201 students had no elevated T scores, that is, *T*-scores ≥ 60T on any of the 22 PAI scales. Table 2 shows the comparable data for the PAI subscales and Table 3 shows the comparable data for the PAI Supplementary Indexes.

Table 4 shows the univariate base rates of elevated scores for the PAI Scales relative to our Canadian samples for ≥ 60T, ≥ 65T, ≥ 70T, ≥ 80T, and ≥ 90T. Multivariate base rates for such elevated or high scores are shown in Table 5.

## DISCUSSION

Canadian university students' PAI mean scale and subscale scores differ substantially from both the US 1990 university student norms and the US 1990 community norms. Relative to the US 1990 adult norms, the differences are large, ranging from approximately −1.0 to +1.8 SD or −10 to +18 T scores. For the PAI scales, the largest differences are seen on ANX (Anxiety), ARD (Anxiety Related Disorders), DEP (Depression), BOR (Borderline

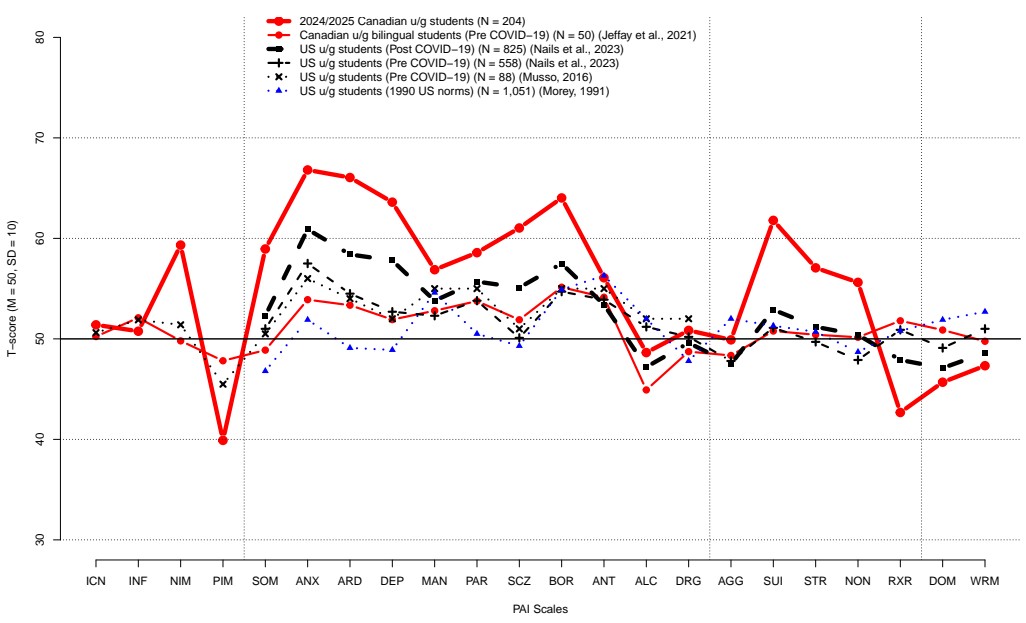

**Figure 3  The PAI scale profile of the mean *T* scores for our Canadian university students and the pro-filed for US and Canadian students reported previously.** The previously reported profiles include the profile of Canadian bilingual university students reported by *Jeffay et al. (2021)*, the 1990 profile of the US college students reported in the PAI manual (*Morey, 2007*), and the profiles of the US college students reported by *Musso et al. (2014)* and by *Nails et al. (2023)*.

Features), SCZ (Schizophrenia), and SUI (Suicidal Ideation) scales. Notably, the Canadian students score high on NIM (Negative Impression Management) but low on PIM (Positive Impression Management) and RXR (Treatment Rejection). Univariate base rates were very high for both elevated ($\geq$ 60T) and high ($\geq$ 70T) scores. Multivariate base rate analyses showed that 95% of Canadian students scored $\geq$ 60T and 73% scored $\geq$ 70T on at least one out of the 22 PAI scales. Thus, when compared to the US 1990 norms, nearly all Canadian university students are abnormal with elevated scores on one or more of the PAI scales.

Relative to the US university student norms, the detailed direct comparison between the US 1990 and the current Canadian student samples (Figs. 3 and 4) reveals shifts towards higher scores on the majority of the PAI scales (*e.g.*, SOM, ANX, ARD, DEP, SCZ, SUI, BOR, PAR, STR, NON), several shifts towards lower scores (*e.g.*, PIM, ALC, RXR, DOM, WRM), and several scales with no or only relatively small shifts (*e.g.*, ICN, INF, MAN, ANT, AGG).

Our results are broadly consistent with a number of other studies showing substantially elevated mean scores on the PAI scales and subscales for samples of both university students and community dwelling adults in the US, Portugal, and Italy detailed in the introduction. We also provide updated norms for the PAI for Canadian university students based on over 200 students in a mid-sized Canadian university including means, standard deviations, McDonald's omegas, Cronbach's alphas, SEms, univariate and multivariate base rates of elevated and high scores (*i.e.*, $\geq$ 60T, $\geq$ 65T, $\geq$ 70T, $\geq$ 80T, $\geq$ 90T).

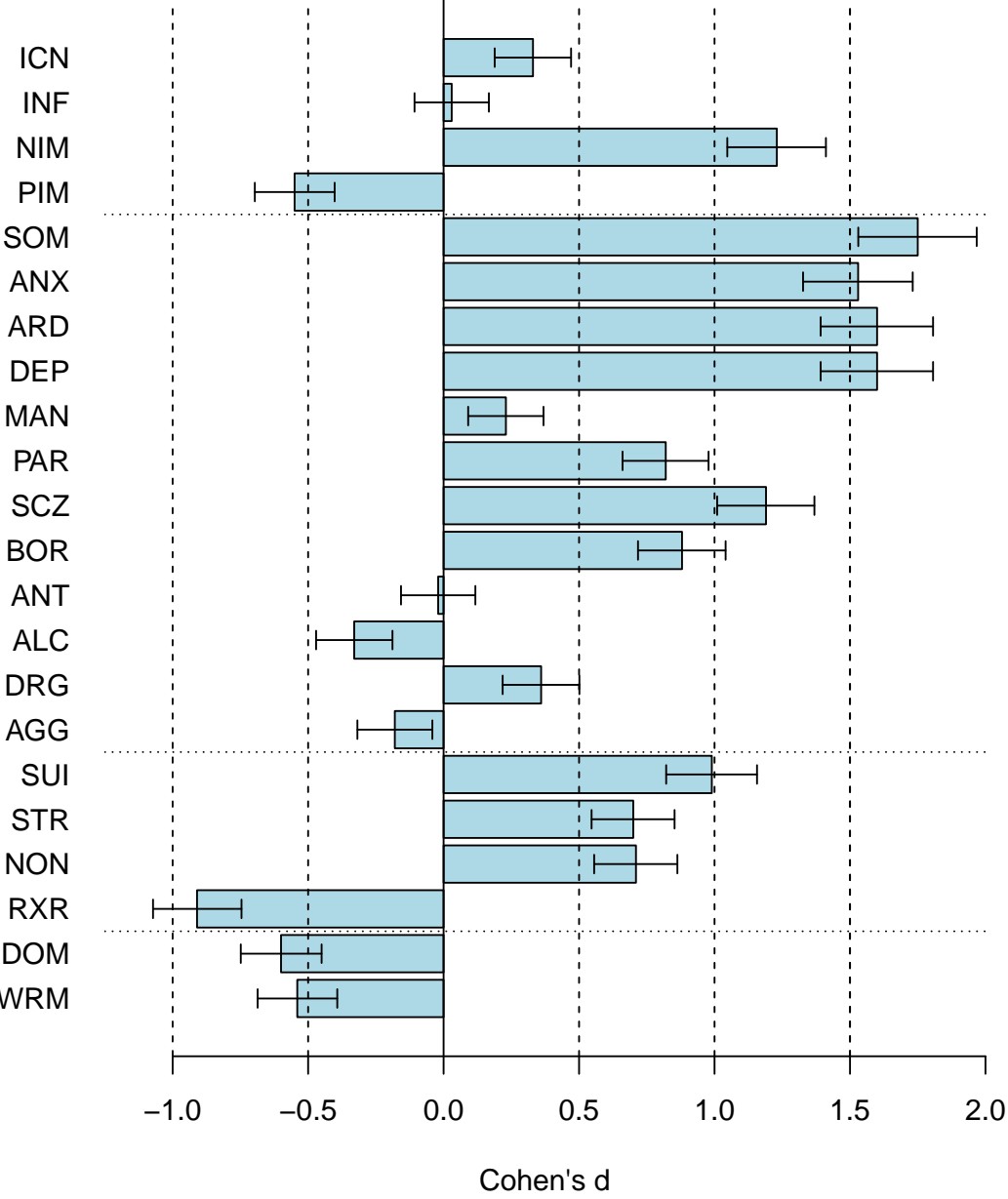

**Figure 4** **The PAI scale profile shifts of the Canadian 2024 *vs.* US 1990 university students in terms of Cohen's d.** The difference between the Canadian 2024 *vs.* US 1990 university students' scores using Cohen's d. Bars denote 95% confidence intervals for Cohen's d.

### Previous "abnormal" has become new normal

The large shifts in how today's Canadian university students score on the PAI are consistent with the previous studies of US university students and with a number of studies recently conducted with both university students and community adults in the US and in other countries, especially post COVID-19 pandemic detailed in the introduction. The large shifts in the PAI scores are also consistent with the wealth of independent evidence demonstrating

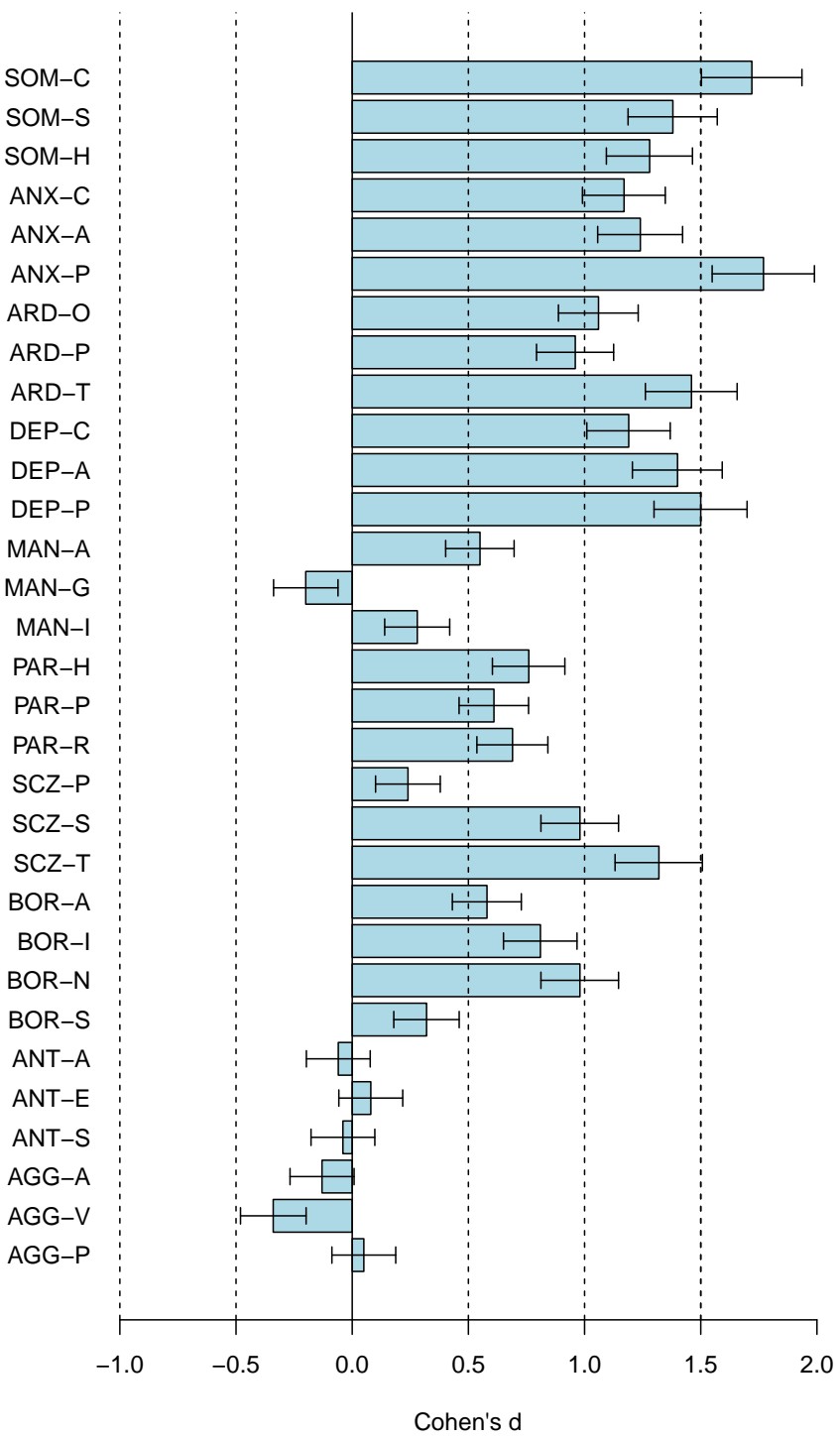

**Figure 5  The PAI subscale profile shifts of the Canadian 2024 *vs.* US 1990 university students in terms of Cohen's d.** The difference between the Canadian 2024 *vs.* US 1990 university students' scores using Cohen's d. Bars denote 95% confidence intervals for Cohen's d.

**Table 3  Descriptive statistics for US 1990 (*Morey, 1991*; *Morey, 2007*) and Canadian 2024/2025 university student samples for PAI Supplemental Indexes.**

| | US | | | | CDN | | | | |
|---|---|---|---|---|---|---|---|---|---|
| | *M* | *SD* | α | $r_{xx}$ | *M* | *SD* | ω | α | *d* |
| Malingering Index (MAL) | 0.58 | 0.74 | n/r | n/r | 0.99 | 0.91 | – | .08 | 0.55 |
| Rogers Discriminant Function (RDF) | −1.01 | 0.83 | n/r | n/r | −0.82 | 0.98 | .01 | – | 0.23 |
| Defensiveness Index (DEF) | 3.04 | 1.71 | n/r | n/r | 2.35 | 1.85 | .61 | .60 | −0.40 |
| Cashel Function (CDF) | 144.09 | 13.29 | n/r | n/r | 148.09 | 15.32 | .05 | – | 0.30 |
| Suicide Potential Index (SPI) | 3.09 | 3.07 | n/r | n/r | 7.75 | 4.73 | .86 | .85 | 1.52 |
| Violence Potential Index (VPI) | 2.31 | 2.40 | n/r | n/r | 3.75 | 3.16 | .76 | .75 | 0.60 |
| Treatment Process Index (TPI) | 1.59 | 2.37 | n/r | n/r | 3.14 | 3.15 | .87 | .86 | 0.65 |

Notes.

ω, MacDonald's ω; α, Cronbach's α; $r_{xx}$, test–retest reliability; *SEm* ($r_{xx}$), Standard Error of Measurement based on $r_{xx}$; SEm (α), Standard Error of Measurement based on α; d, Cohen's *d* with *SD* in denominator based on the US 1990 Norms; n/r, not reported.

US Norms are from *Morey (2007)*: Table 4.15.

large increases in mental health issues and psychopathology, especially in depression, anxiety, and other anxiety related disorders. For example, Statistics Canada reported that the 12-month prevalence of depression, anxiety, and anxiety related disorders increased substantially between 2012 and 2022: nearly doubled for Major Depressive Episode (4.7 *vs.* 7.6%), doubled for Generalized Anxiety Disorder (2.6 to 5.2%), and more than doubled for Social Phobia (3.0 to 7.1%). In contrast, Statistics Canada reported no increase in Alcohol Use Disorder (3.2% to 2.2%), Cannabis Use Disorder (1.3% to 1.4%), and Other Drug Use Disorder (0.7% to 0.5%) over the same time period (*Stephenson, 2023*). Similar doubling of rates of depression have been reported in US university students by *Lipson, Lattie & Eisenberg (2019)* as well as other countries (*Buizza, Bazzoli & Ghilardi, 2022*). Thus, higher scores on anxiety and depression scales are consistent with increases in diagnosed anxiety and depression disorders among university students.

Moreover, some studies suggest that stigma surrounding some common mental disorders, for example, Major Depression has decreased substantially. To illustrate, *Pescosolido et al. (2021)* found that the percentage of US respondents who expressed unwillingness to work closely with, have as a neighbor, socialize with, make friends with, and marry into family of persons with Major Depression all decreased. Similarly, numerous studies have documented increases in the percentage of students asking for and receiving various accommodations due to various disabilities including test anxiety (*Suhr & Johnson, 2022*). Accordingly, one possible reason for increases in the PAI scores is greater willingness of respondents to disclose aspects of their mental challenges independently of the increased documented prevalence of various mental disorders and disabilities.

The large shifts in the PAI scales scores relative to the US 1990 adult as well as college students normative samples are likely due to both temporal/cohort changes and cultural differences between the US and Canada. The temporal/cohort changes likely include changes in psychopathology, changes in Canadian culture itself, and changes in respondents' openness about mental health, that is, reporting style. The world was a very different place in 1990 when the PAI was normed than today. For example, in 1990, the

**Table 4** Univariate base rates of elevated and high scores over 22 PAI Scales in Canadian 2024/2025 university student sample.

|              | ≥60T % | ≥65T % | ≥70T % | ≥80T % | ≥90T % |
|--------------|--------|--------|--------|--------|--------|
| **Validity** |        |        |        |        |        |
| ICN          | 15.2   | 7.4    | 2.5    | 0      | 0      |
| INF          | 21.1   | 11.3   | 5.9    | 0      | 0      |
| NIM          | 15.2   | 11.3   | 5.9    | 1.0    | 0      |
| PIM          | 17.6   | 5.4    | 1.5    | 0      | 0      |
| **Clinical** |        |        |        |        |        |
| SOM          | 17.2   | 9.3    | 3.4    | 1.0    | 0      |
| ANX          | 19.1   | 6.4    | 2.5    | 0      | 0      |
| ARD          | 18.1   | 9.8    | 1.5    | 0      | 0      |
| DEP          | 18.1   | 9.3    | 2.9    | 0      | 0      |
| MAN          | 15.7   | 7.8    | 2.9    | 1.0    | 0      |
| PAR          | 15.7   | 8.8    | 2.5    | 0.5    | 0      |
| SCZ          | 17.2   | 7.8    | 4.4    | 0.5    | 0      |
| BOR          | 16.2   | 6.9    | 1.5    | 0      | 0      |
| ANT          | 15.7   | 9.8    | 5.9    | 0.5    | 0      |
| ALC          | 13.2   | 7.4    | 5.4    | 2.0    | 1.0    |
| DRG          | 14.7   | 8.3    | 6.4    | 2.0    | 0      |
| **Treatment** |       |        |        |        |        |
| AGG          | 16.2   | 11.8   | 7.4    | 0      | 0      |
| SUI          | 16.2   | 9.8    | 6.4    | 0.5    | 0      |
| STR          | 18.6   | 8.8    | 3.4    | 0      | 0      |
| NON          | 15.2   | 9.8    | 2.9    | 1.0    | 0      |
| RXR          | 19.1   | 8.3    | 2.5    | 0      | 0      |
| **Interpersonal** |   |        |        |        |        |
| DOM          | 15.2   | 7.8    | 2      | 0      | 0      |
| WRM          | 18.1   | 6.4    | 1.5    | 0      | 0      |

**Notes.**

$T$-scores are based on our Canadian 2024/2025 sample.

≥60T %, ≥65T %, ≥70T %, ≥80T%, ≥90T %, the percentage of scores equal or exceeding each $T$-score.

Soviet Empire was collapsing, the Cold War was nearly over, and the prospect of nuclear armageddon was reduced. In contrast, today, Russia is in its fourth year of its "Special Military Operation"—otherwise known as the Russian war to occupy Ukraine, the World War III is regularly discussed in the news and social media, and, similarly, the threats of using nuclear arms by Russia appear regularly in the news. The way people obtain information, socialize, communicate, *etc.* has also changed massively with unprecedented changes in technology, such as an introduction of personal computers, world wide web, smart phones, tablets, and social media. Similarly, the composition of undergraduate students has also changed in the intervening 35 years years with a far greater proportion of population attending universities and graduating with university degrees (*Uttl, Violo & Gibson, 2024*). The COVID-19 pandemic has also resulted in unprecedented changes in society and mental health of populations in general. Critically, one has to keep in mind that personality questionnaires such as the PAI are composed of items, that meaning of the items

**Table 5  Multivariate base rates of elevated and high scores over 22 PAI Scales in Canadian 2024/2025 university student sample.**

| No. scores | ≥60T | ≥65T | ≥70T | ≥80T | ≥90T |
|---|---|---|---|---|---|
| 0 | 10.3 | 31.9 | 59.8 | 92.2 | 99.0 |
| ≥1 | 89.7 | 68.1 | 40.2 | 7.8 | 1.0 |
| ≥2 | 77.5 | 43.6 | 21.1 | 1.0 | 0 |
| ≥3 | 52.9 | 28.9 | 8.8 | 0.5 | 0 |
| ≥4 | 40.7 | 17.2 | 8.8 | 0 | 0 |
| ≥5 | 30.4 | 12.3 | 3.4 | 0 | 0 |
| ≥6 | 22.5 | 7.8 | 2.0 | 0 | 0 |
| ≥7 | 16.7 | 5.9 | 1.5 | 0 | 0 |
| ≥8 | 13.2 | 3.4 | 0.5 | 0 | 0 |

Notes.
≥60T %, ≥65T %, ≥70T %, ≥80T %, ≥90T %, the percentage of scores equal or exceeding each $T$-score.

changes, and that the endorsement of items changes with external conditions/situations independently of persons' psychopathologies. For example, one can hardly consider greater endorsement of the PAI items asking respondents how much they worry about things today *vs.* 35 years ago as an indication of greater psychopathology. Critically, regardless of what has caused the changes in the PAI responses, the PAI norms are no longer valid because what once was "abnormal" is now the new normal.

## PAI norms are outdated and obsolete

The PAI and many other psychological tests define what is normal in reference to a bell-shaped distribution of scores obtained by a representative sample of individuals from the populations of interest. Accordingly, when a population's characteristics change over time due to a variety of changes and historical events, for example, due to changes in culture, education attainment, technology, and due to historical events and crises such as COVID-19 pandemic, the previously established norms become outdated, obsolete, and invalid. Accordingly, the tests have to be renormed to capture changes in the population's characteristics and to establish the new norms, the new normal.

The PAI US norms (*Morey, 1991*) are 35 years outdated and obsolete. Our data show that they do not describe Canadian university students and the data we reviewed demonstrate that they no longer describe even the US university students or the US adult non-clinical population. In clinical settings, any interpretations of the PAI scores based on the US outdated norms are likely wrong, misclassifying large proportions of examinees as having elevated or high scores on various clinical and other scales when in fact they are perfectly normal, average, relative to their *current* peers. It is now nearly guaranteed that if one is assessed with the PAI US norms one is likely to have one or more elevated or high PAI scores. For example, using the obsolete PAI norms in fitness for duty assessment, an employee scoring 65T on the Anxiety scale would be flagged as unusually anxious, and possibly unfit for duty even though the employee's Anxiety score was average and no different from other employees.

Accordingly, the PAI outdated US norms ought not to be used in Canada nor elsewhere and ought to be withdrawn by the test developer and the test publisher. The continued use of the US norms has likely caused harm to countless examinees, especially in forensic, employment, fitness-for-duty, child custody, and other high stakes assessment settings where obtaining PAI scores ≥ 60T may result in ineligibility for or loss of employment, loss of child custody, and other serious adverse effects.

Importantly, a number of the PAI validity indexes (*e.g.*, PIM, CDF) featured in the PAI Manual are based on comparing very small groups of examinees asked to complete the PAI under various conditions. For example, the PIM index was developed by asking a small sample of 45 university students "to portray a very favorable self-impression, as if they were attempting to maximally impress a potential employer" (*Morey, 2007*, p. 162). Morey then compared the distribution of the PIM scores of these 45 students to the US 1990 Adult Normal sample and chose to set the PIM cutoff ≥ 57T, "the point at which the actual and simulated distributions cross." (p. 161). Because of the large overlap in the distributions, this 57T cutoff correctly identified 82% of the 45 students as "faking good" and *incorrectly* classified 30% of the US 1990 Adults Normative sample as "faking good". Similarly, the Cashel Discriminant Function (CDF) was developed on a small sample of 44 offenders and then cross-validated on a small sample of 36 university students (*Cashel et al., 1995*). In the Faking condition (23 students), 20 (87.0%) were identified as faking and 3 were identified as honest. In the Honest condition (13 students), 7 (53.8%) were identified as Honest and 6 (46.2%) were identified as "Faking". The CDF is presented in the PAI Manual as "designed to optimally distinguish between defensive *vs.* honest responding" (p. 32) even though it misclassifies nearly 1/2 of honest examinees as "Faking". Thus, these studies are not only tied to the outdated obsolete norms but they are also outdated and obsolete themselves since today's university students respond differently under standard conditions and may also respond differently under the "fake good" conditions. Moreover, they are often based on such small samples that their findings are neither reasonably accurate nor generalizable beyond a study sample itself. No one should be labelled as "faking good" or dishonest in any high stakes assessment based on indexes that are obsolete and misclassify up to nearly 1/2 of the respondents as "faking good".

## Reliability of many PAI scales and subscales is inadequate for high stakes assessments

Previously published data as well as our data also show that the reliability of nearly all of the PAI scales and subscales is inadequate for use in high-stakes assessments. Although reliabilities in the .70 to .80 range may be acceptable for conducting research and reliabilities in .80 to .90 range may be acceptable for using tests in low stakes situations such as vocational counselling, a reliability of .90 is a bare minimum and a reliability over .95 is desirable for using tests in high stakes decisions such as eligibility for insurance benefits, fitness for duty assessments, child custody evaluations, *etc.* (*Kaplan & Saccuzzo, 2013*; *Nunnally & Bernstein, 1994*; *Reynolds, Altman & Allen, 2021*). For example, *Kaplan & Saccuzzo* (*2013*, p. 124-125) wrote:
... It has been suggested that reliability estimates in the range of .70 to .80 are good enough for most purposes in basic research... In clinical settings, high reliability is extremely important. When tests are used to make important decisions about someone's future, evaluators must be certain to minimize any error in classification. Thus, a test with a reliability of .90 might not be good enough. For a test used to make a decision that affects some person's future, evaluators should attempt to find a test with a reliability greater than .95.

As can be seen in Tables 1 and 2, only one out of 44 reliability coefficients reported in the PAI manual (*Morey, 2007*) for the PAI Scales for the college student sample reached the desirable bare minimum of .90 in terms of Cronbach's alpha or test-retest reliability. In our sample, only five out of 22 Cronbach's alpha coefficients reached or exceeded .90. For the PAI subscales, none of the reliabilities reached .90 in the US college sample nor in our sample. Low reliabilities result in large standard errors of measurements (SEm) and wide 95% confidence intervals. Whereas a clinician may think that an examinee's PIM = 57T indicates some attempt to present themselves in a positive light, the 95% CI ranges from 47T to 67T indicating that the clinician has no evidence whatsoever the examinee engaged in any positive impression management.

Disappointingly, the PAI manual (*Morey, 2007*) does not mention the terms "error of measurement" or "confidence interval" even once in "Chapter 2: Administration and Scoring" or "Chapter 3: Interpretation", the two chapters psychologists are most likely to read, if any. In the entire 385 page long manual (*Morey, 2007*), "confidence interval" is mentioned once, on page 135:

> The standard error of measurement, expressed in points on the *T*-score scale, represent the standard deviation of the hypothesized error term as a contribution to observed PAI *T*-scores. Thus, one can be 95% certain that a participant's true score on a PAI scale falls within 1.96 standard errors of measurement of his or her observed score. This confidence interval is particularly useful in attempting to evaluate the meaning of change over time in a person's score–changes that do not exceed this interval are not likely to be clinically meaningful.

Notably, immediately after explaining what the 95% confidence interval is, the manual directs the reader's attention to evaluation of "the meaning of change over time in a person's score" rather than on the more immediate and more serious problem that *every psychologist* ought to know and consider when attempting to interpret test scores: what is the range of true scores that may have produced this observed score? Equally notably, the PAI interpretive software, including the latest version—PAI Plus (published in 2020 but using the original 1990 US norms), gives no indication of the SEms, 95% CIs, and no indication at all that the PAI's scale and subscale scores and indexes are observed scores and merely estimates of the examinees' true scores.

The problem of overpathologizing normal results and ignoring poor reliability of the PAI scales is illustrated by the PAI Manual's exposition of the PIM scale's rather low abnormality cut offs. The manual states (*Morey, 2007*, p. 31):

Moderate elevations (*i.e.,* 57T to 67T) suggest that the individual responded in a manner to portray himself or herself as relatively free of the common shortcomings to which most individuals will admit. This response style could be overt, but it could also involve a covert, automatic defensive process. With PIM in this range, the accuracy of interpretations based on the PAI clinical scales profile may be distorted, and interpretive hypotheses should be reviewed with caution. It is likely that the PAI profile will underrepresent the extent and degree of significant test findings, although the influence on the PAI scales may be scale specific. For example, a respondent may be minimizing a drug problems but accurately describing problems with aggression.

Accordingly, if an examinee scores 57T on the PIM (*i.e.,* 2/3 standard deviation above the mean of the normative sample), a psychologist who takes the manual for its words (and is unaware of what the observed scores actually represent, unaware of SEm, unaware of 95% confidence intervals) may report that the examinee tends "to portray himself as relatively free of the common shortcomings to which most individuals will admit"—copying the phrase directly from the test manual or the PAI computerized report—and decide not to interpret the PAI profiles because of the examinee's "defensiveness." The psychologist may not even bother to quote the statements nor cite the source of these boiler plate, stock phrases lifted directly from the manual or from the computerized report. A reader, an adjudicator, a court, or a tribunal assumes that the lifted phrase was actually the psychologist's original work, presumably arrived to after thorough assessment and after some independent thought about the assessment, its circumstances, *etc.* by the psychologist/assessor. Given no quotation and no citation, the reader, the adjudicator, the court or the tribunal has no indication that the phrase was lifted straight from the manual or from the interpretive report. The examinee is forever labelled as a faker, or even worse, "defensive".

The PAI manual's approach of classifying examinees based on the PIM of 57T as portraying themselves "as relatively free of the common shortcomings to which most individuals will admit" is accusing the examinee of providing dishonest responses or lacking insight into their faults. This accusation is unscientific, inappropriate, unconscionable, and dishonest in and of itself. First, the PIM of 57T classifies 25% of the normative sample as being dishonest or lacking insight by definition, without any evidence whatsoever that they were dishonest or lacking insight.

Second, the SEm of the PIM is approximately 5T, and –quoting the manual's interpretation of the 95% CI—"thus, one can be 95% certain that a participant's true score on a PAI scale [PIM] falls within 1.96 standard errors of measurement of his or her observed score [PIM = 57T]", that is, 47T to 67T. Since 47T is below 50T, the average of the PIM's distribution in normative sample, the psychologist has *no evidence whatsoever* that the examinee's true score is at or above 57T, and thus, no evidence that the examinee is dishonest or lacking insight, no evidence that "the PAI clinical scales profile may be distorted", and no evidence that "it is likely that the PAI profile will underrepresent the extent and degree of significant test findings...".

Third, *Morey (2007)* himself acknowledged the Positive Impression Management does not measure only the Positive Impression Management; Morey wrote (p. 30):

> ... There are a number of reasons why people completing a self-report instrument might not report negative characteristics. One possibility is that the respondent indeed does not have negative characteristics [the instrument asked about], or at least has fewer than most individuals. A second possibility is that they are not telling the truth—that they are trying to deceive... A third possibility is that they are simply not aware of certain faults that they may have... It is these later two characteristics that PIM was designed to measure.

What the PAI "was designed to measure" is irrelevant; the question is what it actually measures. The PIM measures not only the "later two characteristics" but also the first one, and thus, the PIM's interpretation offered in the manual is completely unreasonable, and ultimately causing harm. *Morey (2007)* himself admits that a large proportion of the normal population is identified as "faking good" or being dishonest by indexes such as the PIM (p. 30):

> It should be recognized that the tendency for favorable self-presentation appears to be fairly common in the normal population. Typically, most cutoff scores on indexes of social desirability that were derived from clinical studies will identify 30% to 40% of the general population as "faking good." Such results underscore the difficulty of distinguishing defensive responding from normality with respect to clinical instruments...

If the index identifies honest examinees as "faking good", the index is wrong, inappropriate, and its use in any high or even low stakes assessments is unconscionable and unethical. Labelling an examinee as dishonest, as "faking good", is a severely defamatory accusation. No psychologist should ever make a claim that the examinee is dishonest when the psychologist has no idea what caused the particular score, has no idea about the errors of measurement, and has no idea how to interpret psychological scores. The severity of this relevant non-disclosure by psychologists is further compounded by the fact that a reader of a psychological report who is not a psychologist and not an expert (a) has no indication that they were misled, and (b) has no way to access the test manual to find out that they were misled about the PIM since only qualified psychologists can access the PAI manual but the public can not.

One may argue that an examinee has no need to access the PAI manual because the examinee can request that the examinee's PAI data and materials be sent to another psychologist for a review or the examinee can apply to the Court for access to the raw scores, interpretive reports and even the PAI manual. This argument ignores the fact that neither option is accessible to ordinary people who do not have extra $5,000 to $10,000 to hire another psychologist or a lawyer to represent them in the court system to obtain access to their test scores, interpretive reports and the test manual.

## PAI multivariate base rates of elevated/abnormal scores have not been reported

The PAI manual, the PAI computerized interpretive reports, and the PAI training materials are also silent on the problem of multivariate base rates of abnormal scores, that is, percentage of examinees within the normative samples with one or more elevated scores across 22 primary scales. When healthy normal people complete a battery of tests or a test with multiple measures such as the PAI, a large proportion of them will obtain one or more "abnormal" or "elevated" scores. When one focuses on one measure or one scale in isolation, a cutoff for elevated scores set at 60 T will identify approximately 16% of normative sample as having elevated scores. However, when one applies this same cut off to 22 primary scales simultaneously, a large proportion of the normative sample will have one or more elevated scores. If one rolls a die once, a probability of getting 1 is only about .17. However, more times one rolls a die, greater the probability that one rolls 1 at least once. Rolling a die 22 times is likely to result in at least one 1 with probability nearly 1.00. Accordingly, when a clinical psychologist administers the PAI and looks for elevated scores, the clinician is likely to find at least one elevated score among the 22 primary scales. To put it bluntly, an ignorant or unscrupulous forensic psychologist has 22 ways to find some problematic psychopathology in each examinee based in part on random chance due to low reliability of the PAI scales and large number of measures considered. Moreover, the clinician will find more elevated scores the lower "elevation" cut off the clinician chooses.

Although there appears to be no awareness of this problem in the personality assessment literature, the problem of the multivariate base rates has been recognized, acknowledged, and widely discussed in intelligence testing for decades (*Crawford, Garthwaite & Gault, 2007*; *Karr et al., 2017*; *Karr et al., 2018*; *Karr et al., 2020*). For example, in one of the extensive reviews of the problems of multivariate base rates, *Brooks, Iverson & Holdnack (2013)* identified five key psychometric principles that need to be considered when interpreting multiple test scores on intelligence tests. The first three of those principles apply, by definition, to any kind of assessments including personality assessments: (1) abnormal (*e.g.,* low or high) scores are common across multiple measures, (2) the number of abnormal scores depend on the cutoff score used, (3) the number of abnormal scores increases with the number of scores considered.

Our research shows that using liberal cutoff for abnormality (60T or 84th percentile) and outdated obsolete US 1990 norms identifies nearly all of our students (95%) as "abnormal", having one or more elevated scores across the 22 primary scales. Similarly, using the same liberal cutoff for abnormality (60T or 84th percentile) with T scores calculated from within our Canadian sample (*i.e.,* not relative to the US norms) for each of the 22 PAI primary scales, only 10.3% of our sample showed no elevated scores, nearly 90% had one or more "abnormal" scores, and over 1/2 had 3 or more abnormal scores. Thus, the presence of a number of elevated scores is normal and not indicative of any kind of abnormality. Moreover, when abnormal scores are defined by deviation from the average, presence of abnormal scores across multiple measures is mathematical necessity. Accordingly, any interpretation of abnormal scores must take into account prevalence of abnormal scores in normal healthy normative samples. Clinicians who do not consider multivariate base rates

are likely to misclassify large number of examinees as having some kind of psychopathology when they do not.

## First, do no harm! Ethics of using outdated, obsolete and invalid tests

The Standards for Educational and Psychological Testing (*American Educational Research Association, American Psychological Association & National Council on Measurement in Education, 2014*) are clear that it is test users' responsibility to periodically verify that continued use of tests, norms, and data is still appropriate, and explains in a commentary why this periodic verification is necessary. The Standard 9.7 states:

> Test users should verify periodically that their interpretations of test data continue to be appropriate, given any significant changes in the population of test takers, the mode(s) of test administration, or the purposes in testing.

> **Comment:** Over time, a gradual change in the characteristics of an examinee population may significantly affect the accuracy of inferences drawn from group averages. Modifications in test administration in response to unforeseen circumstances also may affect interpretations.

The American Psychological Association's Ethical Principles of Psychologists and Code of Conduct (*American Psychological Association, 2017*) includes several relevant principles that bar psychologists from using outdated and obsolete tests and from using tests whose norms, validity and reliability were not established for the target population. The Principle 9.08 states:

> **9.08 Obsolete Tests and Outdated Test Results**
> (a) Psychologists do not base their assessment or intervention decisions or recommendations on data or test results that are outdated for the current purpose.
> (b) Psychologists do not base such decisions or recommendations on tests and measures that are obsolete and not useful for the current purpose.
> The principle 9.02 (a) and (b) states:

> **9.02 Use of Assessments**
> (a) Psychologists administer, adapt, score, interpret, or use assessment techniques, interviews, tests, or instruments in a manner and for purposes that are appropriate in light of the research on or evidence of the usefulness and proper application of the techniques.
> (b) Psychologists use assessment instruments whose validity and reliability have been established for use with members of the population tested. When such validity or reliability has not been established, psychologists describe the strengths and limitations of test results and interpretation.

Unfortunately, the APA's and other code of ethics provide no guidelines on how to determine whether tests "are obsolete and not useful for the current purpose"; it presumes psychologists know that tests are obsolete when, for example, norms are no longer valid, test items are outdated, and continued use of tests would lead to incorrect decisions and misdiagnoses. The International Test Commission's Guidelines for Practitioner Use of Test
Revision, Obsolete Tests, and Test Disposal (*International Test Commission, 2015*) come closest in defining test obsolescence, at least "generally speaking":

> ... Generally speaking, a test is obsolete when its underlying theory, item content, **norms** [emphasis added], or technical adequacy no longer meet the needs for its intended purpose, professional standards, or when its continued use would lead to inappropriate or inaccurate decisions or diagnoses.

In contrast, other standards and ethics codes are silent and do not even mention the problems of outdated and obsolete tests. Moreover, more general dicta that psychologists ought to be familiar with and rely on *current* science are often ignored, not only by psychologists but also by some of the regulatory bodies, members of the Association of State and Provincial Psychology Boards (ASPPB) (*Uttl, Violo & Gibson, 2024*; *Uttl, Sikma & Tat, 2025*). For example, the College of Alberta Psychologists (http://www.cap.ab.ca/) considers use of even 80 year old tests, norms, and data as well as comparison of an examinee's scores on a new edition of the test (WAIS-IV CDN) (*Wechsler, 2008*) to norms for the earlier (53 years earlier) edition of the same named test (WAIS) (*Wechsler, 1955*) as meeting their minimum professional standards (*Uttl, Violo & Gibson, 2024*), even though such uses amount to the practise of junk science by psychologists (*Lilienfeld, Lynn & Lohr, 2014*; *Uttl, Sikma & Tat, 2025*). In contrast, the North Carolina Psychology Board considered such uses inappropriate and unprofessional conduct, with an appropriate penalty requiring a psychologist to relinquish their license to practice (*North Carolina Psychology Board, 2022a*; *North Carolina Psychology Board, 2008*; *North Carolina Psychology Board, 2018*; *North Carolina Psychology Board, 2022b*). The North Carolina Psychology Board even issued an official advisory on "Using Current Tests and Norms" (*North Carolina Psychology Board, 2018*) and another reminder four years later (*North Carolina Psychology Board, 2022b*).

Psychologists' ignorance of general dicta to use current, established science to support their clinical opinions and to avoid obsolete and outdated tests, norms, and data has been causing widespread harm to examinees and to the public who incorrectly believe that professionals use current scientifically based methods to assess personality and other psychological constructs (*Uttl, Violo & Gibson, 2024*). Examinees, readers of psychological assessment reports, adjudicators, judges, and the public have no idea that the regulated professionals are routinely using decades outdated, obsolete, and irrelevant tests, norms, and data even in high stakes assessments, and that at least some psychology boards condone such unscientific practices and consider them fine minimally competent conduct (*Uttl, Violo & Gibson, 2024*; *Uttl, Sikma & Tat, 2025*).

With respect to the PAI (and other widely used psychological tests), the problem is compounded by failure to recognize low reliabilities of the PAI scales and subscales, failure to present 95% confidence intervals for the PAI scores, failure to advise users on the folly of interpreting obtained scores as if they were true scores, and failure to take into account multivariate base rates of low scores.

Various reviews of the PAI, often authored or co-authored by researchers with financial stakes and other conflicts of interest in the PAI's sales and success, continue to exalt the PAI

as a modern reliable and validated instrument widely used in forensic and other settings. These same reviewers do not mention massive shifts in what is normal today *vs.* what was normal 35 years ago and captured at that time by the US outdated and obsolete norms. For example, Paulino and his colleagues (*Paulino et al., 2024b*) exalted the qualities of the PAI in various superlatives. To illustrate, the first sentences of paragraphs in their discussion section start as follows: "The PAI is a comprehensive, objective, and direct measure that allows for the evaluation of several characteristics of personality with special utility in forensic settings...", "Its applicability in offender and inmate populations is of particular interest...", "In other forensic settings, such as intimate partner violence cases, it is also a powerful tool...", "The PAI is also a useful measure in family law cases, where it is mainly used in the evaluation of positive distortion responding...", "The PAI is a good predictor of misconduct, insubordination, and abuse of power in police officers...", and the final paragraph reads: "Overall, the utility of the PAI in forensic settings is generally accepted and its growing recognition is evident in the exponential dissemination to other countries that, in the last couple of years, have adapted it to their populations and forensic realities."

However, *Paulino et al. (2024b)* did not mention low reliabilities, did not mention the wide confidence intervals for the PAI scales including the PIM scale, did not mention the norms are 35 years outdated and obsolete, and did not mention any of the research demonstrating that the normative responses have shifted substantially in the US as well as elsewhere. This blindness to the PAI's fatal flaws may be explained by the competing interests of several of the authors. The declaration of competing interest states that "Dr. Edens is the lead author of the Personality Assessment Inventory Interpretive Report for Correctional Settings (PAI-CS) and receives royalties from its publisher, Psychological Assessment Resources, Inc... The remaining authors declare that they have no known competing financial interests or personal relationships that could have appeared to influence the work reported in this paper." (p. 8). However, the Conflict of Interest declaration in a *different* article (*Paulino et al., 2024a*) states that Paulino, Moniz, Moura, Rijo, Simoes (*i.e.,* every author except Leslie Morey) are "part of the team responsible for adapting the Personality Assessment Inventory, in Portugal" conducted by the PAI's Portugal's publisher Hogrefe Publishing "which approved the current adaptation study (*e.g.,* translation, back translation, preliminary version, and normative data)." It appears that, contrary to the declaration in Paulino, Edens at al., the authors (a) have had "personal relationships" with their co-author, Leslie Morey, who "is the author of the Personality Assessment Inventory and receives royalties from its sale." (p. 14), and (b) were part of the team adapting the PAI to a Portuguese population.

## Limitations

Our data examined the PAI performance of university students only and in one Canadian university only. However, our data are consistent with the large shifts in the PAI performance of the US college students, and these changes are even more pronounced in our sample. Accordingly, although it is clear that the US norms (*Morey, 1991*) are outdated, obsolete, irrelevant, and clearly inappropriate for use with both the US and

Canadian university students today, our data do not constitute new norms for a non-clinical Canadian general population nor for clinical Canadian samples. Our data can be used as norms only for Canadian university students and with awareness and explicit acknowledgement that (a) our data were obtained only in one university and (b) various scales and indexes may no longer be valid measures of constructs they were intended to measure since we have not examined the validity of the PAI scales.

In 1990, the US college students norms were similar to the US census matched norms, especially to the US census matched young adults, but it is unclear whether this similarity continued and whether this similarity would be observed in Canada. Comparison of *Nails et al. (2023)* and *McCredie & Morey*'s (*2019*) data suggest that both the US university students and the US community adult samples score much higher than the PAI 1990 US norms, and that the PAI US norms are outdated, obsolete and invalid for both the college and the census matched samples. However, no comparable studies exist in Canada and it is unclear whether Canadian norms for the university students would differ from those for normal adults. Furthermore, it is unclear whether the PAI scales, subscales, and various supplementary indexes developed using data gathered decades ago are still valid measures of what they were intended to measure.

In any case, the PAI 1990 US norms ought not to be used with any population until researchers establish *current* norms that accurately describe what is normal in the current US as well as the Canadian population. Comparison between *Nails et al. (2023)* and our data suggests that applicability of the US data to the Canadian populations cannot be assumed but must be empirically established.

## Recommendations

The problem of psychologists using outdated and obsolete tests and norms as well as norms developed in another country or culture is not limited to the PAI; it is widespread (*Uttl, Sikma & Tat, 2025*). For example, an alternative to the PAI, Minnesota Multiphasic Personality Inventory- 3 (MMPI-3) also suffers from lack of Canadian norms, validation in Canada, as well as many other issues (*e.g.*, low reliabilities of validity scales, inadequate reliabilities of substantive scales with 95% confidence intervals for most of the scales spanning approximately 20T scores), although its US norms are more recent, collected in 2017 to 2018, in pre-COVID-19 era (*Ben-Porath & Tellegen, 2020*). As another example, the D-KEFS (Delis-Kaplan Executive Function System) popular test battery at least among neuropsychologists was normed on US sample sometime prior to 2001 (the test manual is silent on when the normative sample was actually tested), that is, nearly 25 years ago or more (*Delis, Kaplan & Kramer, 2001*). Just like the PAI, the D-KEFS has never been normed on Canadians. Moreover, the D-KEFS has been heavily criticized for decades for having substandard inadequate reliabilities ranging from .13 (Design Fluency Test, Condition 3: Correct Designs Switching, Ages 8–19, p. 26) and with reliabilities .90 or higher being rare (*Crawford, Sutherland & Garthwaite, 2008*; *Suchy & Brothers, 2022*). Yet, *Monette et al.*'s (*2023*) survey shows that the D-KEFS is used by 82.6% of Canadian neuropsychologists even though it is obsolete, developed and normed in another country, and lacks reliabilities necessary to support clinical decisions. More generally, *Monette et al.*

*(2023)* also reported that 80% of Canadian neuropsychologists responded that they use "normative data from another country", 55.5% responded that one of the challenges was that tests "lack normative data for my country", 29.1% responded that tests were "not adapted to my cultural setting", and 28.3 noted that tests "do not have good psychometric properties."

What can be done to ensure that psychologists do not use outdated obsolete tests and norms? One of the obvious recommendations arising from ours as well as from the previous studies with the US adult and US university students is for the author and the publisher of the PAI to withdraw the PAI and to issue the public advisory that the PAI test score interpretations are no longer valid with neither US nor Canadian populations. According to the Standards for Educational and Psychological Testing (*American Educational Research Association, American Psychological Association & National Council on Measurement in Education, 2014*) these are test authors' and publishers' obligations. For example, the Standard 4.24:

> Test specifications should be amended or revised when new research data, significant changes in the domain represented, or newly recommended conditions of test use may reduce the validity of test score interpretations. Although a test that remains useful need not be withdrawn or revised simply because of the passage of time, test developers and test publishers are responsible for monitoring changing conditions and for amending, revising, or withdrawing the test as indicated.

However, despite the published research (some co-authored by the PAI author himself) showing that both the US adult and US college norms are outdated, obsolete, and no longer valid (*Maffly-Kipp & Morey, 2024*; *McCredie & Morey, 2019*; *Nails et al., 2023*), Morey and the test publisher continue to inform the public and psychologists that the PAI 1990 adult norms continue to be valid. For example, *Morey (2021)* informed attendees of a workshop at the 2021 meeting of the Society for Personality Assessment that the 1990 norms were "pretty much holding up"; Morey stated:

> The PAI, you should know this but make sure you do, the PAI is standardized using a linear transformation to have a mean of 50 and a standard deviation of 10 in the US community sample that was basically, um, census matched on age, race, and gender, um, when the PAI was developed. And, uh, ah, by the way, we've, we've been continuing to do research on those norms, including looking at, uh, online samples using Amazon MTurk and things like that and published, um, recently that really suggests that those norms are pretty much holding up. It's not as if the US is really changing, um, with respect to a lot of the questions the PAI is asking. Those norms are holding up pretty well.

The four hour long recordings of Morey's workshop has been posted on the PAR Inc. (parinc.com) website on August 23, 2022 as one of the PAI training materials accessible to psychologists with the PAR Inc. account (*Morey, 2021*). It appears that the PAI author and the publisher continue to believe that the US 1990 adult/community norms continue to be valid, and that today's US adults are responding the same way on the PAI even though this

belief is objectively false and demonstrated to be false by the PAI's author's own research and numerous other studies. This belief may not be surprising given that test publishers are corporations established to make profits and that test authors collect royalties from tests and related merchandise sales. Accordingly, test authors and test publishers are unlikely to periodically re-examine continued reliability, validity, and currency of their tests, and to renorm them unless they are forced to do so by external forces, for example, overwhelming research demonstrations that their tests are no longer valid for their stated purposes, test users refusals to buy and to use their outdated and obsolete tests, and test information consumers (*e.g.*, employers, courts) refusing to consider the PAI assessment results.

Although ethics codes make test users, for example, clinical, vocational, and forensic psychologists, ultimately responsible for using current, reliable, and valid tests, test users often obtain their information about tests from their graduate school training, test manuals, and from test publishers' sponsored talks, training videos, and webinars such as the one noted above. Most clinical psychologists do not do relevant research and are unlikely to go and read primary sources (often published behind paywalls) and examine scientific evidence of test norm currency and test reliability and validity for themselves. One way forward would be to require all clinical psychologists who wish to conduct formal assessment using psychological tests to be periodically required to pass an exam on reliability, validity, currency and interpretation of psychological tests including exams on reliability, validity and currency of specific tests such as the PAI. Such requirements would likely be very unpopular among clinical psychologists and other test users.

Yet, importantly, clinical psychologists have a duty to obtain "informed consent" for assessment from all examinees. As detailed in one ethics code, Canadian Code of Ethics for Psychologists, this requires psychologists to do the following (Principle I.23) (*Canadian Psychological Association, 2017*):

> Provide, in obtaining informed consent, as much information as reasonable or prudent individuals and groups .... would want to know before making a decision or consenting to the activity. Typically,…, this would include: purpose and nature of the activity; … likely risks and benefits of the activity...

Thus, an informed consent form for administration of the PAI to Canadian adult as part of a fitness for duty assessment ought to disclose to an examinee that they would be tested with the 35 years old tests, that their scores will be compared to scores of examinees tested in a different country 35 years ago, that a psychologist's opinion of the examinee's personality based on the PAI will have no scientific basis, and that one risk of this activity is that the examinee may lose their job for no reason whatsoever. We propose the following text to be included in an informed consent for assessment of adults in Canada with the PAI at this time:

> I will administer to you a personality questionnaire called the Personality Assessment Inventory (*Morey, 1991*; *Morey, 2007*). To interpret your scores, I will compare them to scores of US examinees tested 35 years ago in several selected US states because no one has established how Canadians such as yourself score on this test. To be clear, although

I will use performance of US examinees 35 years ago, I have no scientific evidence how people such as yourself score on this test today here in Canada. Accordingly, my opinions about your personality will have no scientific basis but they may be used by your employer to terminate your employment. Please initial here __ to indicate your informed consent for me to interpret your scores using the 35 years old test norms from a different country.

We believe that no adult examinee competent to give informed consent would consent to such an unscientific assessment if these facts were disclosed to them in the informed consent form prior to the assessment. Ethics codes and standards of practice ought to make these disclosures—tests to be used, their age, origins of test norms, *etc.*—mandatory, in writing, as part of informed consent process. Examinees have a right to know this essential information prior to proceeding with the assessment.

Another recommendation is for the regulatory bodies established to protect the public (in the US and Canada provincial and state psychology boards) to do what the test publishers ought to do and to set up standing committees to be "responsible for monitoring changing conditions" and issuing the public advisories on currency, reliability, and validity of various psychological tests, especially on tests most frequently used in high stakes decisions such as employment selection, fitness for duty assessment and child custody evaluations. The problem with this recommendation is that the regulatory bodies are not necessarily any more knowledgeable than an average clinical psychologist who obtains their knowledge about tests' currency, reliability, validity, and wider scientific underpinning, if any, from test publishers and test publishers' sponsored talks, training videos, and webinars such as the one the quote about the PAI's 1990 adult norms continued validity comes from. As detailed above, the College of Alberta Psychologists considers use of even 80 years old intelligence tests, norms, and data at least minimally competent practice despite massive amounts of research evidence (some summarized in test manuals themselves) that such use of 80 years obsolete intelligence tests and data amounts to practice of junk science (*Uttl, Violo & Gibson, 2024*).

Apparently, some psychologists believe, incorrectly, that tests and norms are valid forever or until someone demonstrates that the tests and norms are invalid. For example, *Russell* (*2010*, p. 66) wrote:

... neuropsychology should recognize that the research, including norming, based on a validated test or battery remains valid until it has been demonstrated to be invalid. Regardless of how long ago a validated test was published, its results are still sound unless research has demonstrated its lack of validity.

This view is wrong and misguided (*Uttl, Sikma & Tat, 2025*); it *assumes* that the world, populations, cultures, education, technology, climate, *etc.* are not changing but remain static. Since the world, population, cultures, *etc.* change, and sometimes very rapidly (*e.g.,* COVID-19 pandemic), one cannot assume that tests and norms remain valid but one has to assume that tests and norms become invalid not because of simple passage of time per se but because of the changes in the world, populations, *etc.* that occur with time. Thus,

reliability, validity, and currency of tests and norms have to be periodically re-examined to determine whether or not changes in the world, populations, *etc.* changed how *current* samples from those populations perform on psychological tests including how they answer questions on personality tests such as the PAI *today vs.* 35 years ago, or before *vs.* after extraordinary events such as pandemics.

Critically, reliability, validity, and *currency* of psychological tests needs to be periodically verified and re-established by researchers with no conflict of interest, and no financial and reputational benefits tied to outcomes of such research. Any such research ought to be transparent, published in open access journals, published with accompanying data, and widely available to the public including to examinees affected by use of these tests, their lawyers, and other consumers of the PAI assessments and opinions based on those assessments. Establishing reliability, validity, and currency of psychological tests in original as well as new contexts (*e.g.*, countries, cultures, situations) are projects imminently suitable for undergraduate honours as well as Master's theses projects. When such research yields evidence of poor reliability, lack of validity, and/or shifting norms, it would be incumbent on researchers to notify regulatory bodies as well as provincial, state, and national psychological associations to issue advisories to its members alerting them to new data on reliability, validity, and currency of such psychologist tests and norms. If the College of Alberta Psychologists' acceptance of 80 years obsolete tests and norms as minimally competent ethical practice is any guideline, regulatory bodies and their members themselves may not become aware of outdated nature of many tests and norms as well as resulting harm for centuries to come if researchers themselves do not take time to alert them.

Finally, as a rule of thumb, we suggest that if a test or test norms are older than 10 years, test users ought to investigate scientific evidence of their continued reliability, validity and currency prior to using the test and/or norms. If the users *cannot find and cite* scientific evidence of continued reliability, validity and currency, the users ought to stop using the test and norms to avoid harming examinees and society, especially in high stakes, life altering assessments. We note that a widespread use of a test and norms is *not* the scientific evidence of the test and norms' reliability, validity and currency. Only scientifically sound research studies, with adequate samples representing populations of interest, can establish continued reliability, validity and currency of tests and norms in particular contexts (*e.g.*, Canada *vs.* US, younger *vs.* older examinees).

## CONCLUSIONS

The PAI's US norms are antiquated, and clearly invalid for Canadian university students, US university students, and US community dwelling adults. Using the PAI's outdated US norms in a Canadian context is practising junk science and likely amounts to malpractice. While it may appear scientific to interpret the PAI scores with reference to *some* norms, when those norms are outdated, obsolete, developed in a different country and culture, and otherwise invalid the interpretation is not based on current science but on junk science (*Lilienfeld, Lynn & Lohr, 2014*, pp. 1–12, 282–283). Equally problematic are the PAI's low

reliabilities. Although some of the PAI's reliabilities are adequate for research, the PAI reliabilities are not adequate for using the PAI in high stakes testing with life altering consequences, for example, in forensic settings such as child custody, employment, and fitness for duty assessments. The PAI's use in high stakes assessments amounts to forming opinions about individuals based in large part on random error and thus, engaging in junk science practice.

The PAI users ought to focus on first doing no harm. To avoid doing harm, psychologists need to have necessary understanding of tests, read test manuals, understand test norms, understand test reliability, and understand inferences that can be reasonably drawn from observed test scores, in other words, they must also be knowledgeable in psychometric theory and data interpretations. When the test norms are (a) from a different country/culture, (b) based on a sample from a different population, and/or (c) outdated, obsolete and invalid, psychologists ought not to use these invalid and/or inappropriate test norms until such time as current scientific data become available that establish new test norms, including reliability and validity of the test for the populations of interest. In case of the PAI, psychologists ought to abandon the US norms and the test developer and the test publisher ought to withdraw the US norms immediately to avoid further harm to examinees and society.

Given the PAI's widespread use, it is likely that thousands of examinees have been harmed and continue to be harmed by psychologists' use of the PAI outdated, obsolete, and invalid norms. A person sent to a psychologist for an assessment with the PAI and US norms *today* is virtually guaranteed to have one or more elevated scores and one or more "psychopathologies" simply due to outdated norms and psychologists' lack of familiarity with essential psychometric concepts such as the difference between observed *vs.* true scores, reliability, 95% confidence intervals, and univariate and multivariate base rates of "abnormal" scores.

## ACKNOWLEDGEMENTS

Preliminary results of this study were presented at the Canadian Society for Brain, Behavior, and Cognitive Science meeting at Edmonton, AB, Canada, June 25–27, 2024.

### Funding

The study was funded by Natural Science and Engineering Research Council (NSERC) Discovery Grant to Bob Uttl. The funder had no role in the design, data collection, data analysis, decision to publish, preparation of the manuscript, and reporting of this study. The funders had no role in study design, data collection and analysis, decision to publish, or preparation of the manuscript.

### Grant Disclosures

The following grant information was disclosed by the authors:
Natural Science and Engineering Research Council (NSERC).

## Competing Interests

The authors declare there are no competing interests.

## Author Contributions

- Bob Uttl conceived and designed the experiments, performed the experiments, analyzed the data, prepared figures and/or tables, authored or reviewed drafts of the article, and approved the final draft.
- Kiefer Sikma performed the experiments, prepared figures and/or tables, authored or reviewed drafts of the article, and approved the final draft.
- Mikayla Tat performed the experiments, prepared figures and/or tables, authored or reviewed drafts of the article, and approved the final draft.

## Human Ethics

The following information was supplied relating to ethical approvals (i.e., approving body and any reference numbers):

The Mount Royal University Human Research Ethics Board (HREB) granted Ethical approval to carry out the study (HREB File #103697).

## Data Availability

The data is available at OSF: Uttl, Bob. 2025. "Personality Assessment Inventory." OSF. June 2. doi: 10.17605/OSF.IO/V62YR.

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
