# Peer review of "Personality Assessment Inventory (PAI): obsolete norms identify psychopathology in nearly everyone"

_PeerJ, doi:10.7717/peerj.20340_

## Round 0.1 · original submission · Major Revisions

· Academic Editor

Major Revisions

This paper is interesting and potentially important - but needs major revisions in order to attain its full impact. Please go over both reviews, which are positive and detailed. Please address all the comments of both reviewers so that your work is easy to follow. The most important methodological point made by both reviewers has to do with the sampling for creating norms. I believe this is a general problem with well used psychological tests including intelligence tests. You may want to bring this up in the Discussion and suggest some ground rules for creating clinical cut-off points and norms. As stated in the APA ethical principles "Standard 9.08, Obsolete Tests and Outdated Test Results, of the APA Ethical Principles of Psychologists and Code of Conduct, states that psychologists do not base such decisions or recommendations on tests and measures that are obsolete and not useful for the current purpose. However, no guidance is provided on how to determine when a test is obsolete"

Please note that the original norming sample and the one you provide are college samples and may not be relevant to the current older adult distressed individuals who are administered the PAI for placement decisions.

Reviewer 1 ·

Basic reporting

First of all, I would like to thank you for the opportunity to review this article.

As far as I understand, the authors examine whether the PAI’s normative data, established in the early 1990s in the U.S. population, still accurately differentiate normal from pathological personality profiles today. The authors administered the PAI to 207 Canadian undergraduate students, finding that mean scores on many clinical scales (e.g. Anxiety, Depression, Borderline, Schizophrenia) were in the “moderately elevated” range (T-scores around 60–69) relative to the old U.S. norms. Moreover, about 95% of these students had at least one PAI scale ≥60T (i.e., elevated) and 73% had at least one ≥70T (i.e., high). In other words, nearly every student would appear to have some form of psychopathology if interpreted by the original norms. The authors also report that several PAI scales and indices showed suboptimal internal consistencies (Cronbach’s α), adequate for research but “too low” for high-stakes clinical decisions. Given these findings, the authors conclude that PAI’s 35-year-old U.S. norms are obsolete and should not be used for high-stakes assessments. They argue that continuing to use the outdated norms (e.g. in forensic evaluations, employment screenings, custody disputes) risks mislabeling normal individuals as disordered, which they even liken to professional malpractice if uncorrected.

Experimental design

Study Design and Sample Representativeness.
The core argument is based on the analyses using a sample of 207 Canadian university students, a convenient and relevant group to test the authors’ hypothesis (i.e., young adults in 2020s vs. norms from 1990). The data collection appears straightforward and well-documented, and focusing on a student sample is reasonable given prior suggestions that college populations today score higher on PAI scales than before. However, the generalizability of this sample is limited, as all participants were undergraduate psychology students, most likely in their late teens or early 20s (M = 21.2 years, SD = 4.96). While the authors extrapolate their conclusions to the broader “normal adult US population,” this inference is speculative without a more diverse sample. It remains plausible that non-student adults, particularly older, working individuals, have not experienced comparable shifts on certain PAI scales. Notably, college students, and psychology students in particular, have been shown to report elevated levels of psychological distress relative to the general population (Bayram & Bilgel, 2008; Ibrahim et al., 2013). This matters because if the observed over-pathologizing is partly driven by sample characteristics, the implications for the general adult population, and especially for forensic or clinical applications, may be overstated.

The manuscript would be stronger if the authors acknowledged this demographic limitation and, if available, integrated any evidence on PAI trends in non-college adult samples. For example, if published data or pilot results exist for community or older adult samples, discussing those would help justify the claim that all U.S. norms, not just college norms, are outdated. If such data do not exist, the authors might adapt their language to focus on the clear discrepancy in college students, while suggesting that similar norm shifts may be occurring in other populations. This distinction is important, because high-stakes PAI use often involves diverse adults (e.g., job applicants, parents in custody cases, etc.), not only college-aged individuals.

Additionally, cultural differences should be considered. As stated by the authors, the PAI has never been normed on a Canadian sample, and the authors appropriately note that users simply assumed U.S. norms apply to Canadians. The results indeed suggest Canadian students score higher than the U.S. norms, but is this entirely due to outdated norms (temporal change), or could there be a consistent Canada–US difference? The authors compare their Canadian sample not only to 1990 U.S. college norms but also to a recent U.S. student sample (Nails et al., 2023). They report the Canadian students’ profile was elevated relative to both, which suggests Canadian young adults today are similar to or even more elevated than U.S. young adults today. Still, providing the exact comparisons (e.g. mean differences between Canadian vs. post-2010 U.S. samples) would help readers judge how much temporal change vs. cultural difference is at play. In any case, it would be wise for the authors to recommend local re-norming (for Canada and other countries) in addition to updating the U.S. norms, since psychological test norms often do not generalize perfectly across cultures (e.g., Dong & Dumas, 2020; Schmitt et al., 2007). Measurement non-invariance is widespread. Systematic reviews show few personality measures meet basic equivalence requirements across countries, this justifies local norming for accurate interpretation. Aggregate trait differences across hundreds of thousands of respondents confirm that profile norms shift by region.

Validity of the findings

Statistical Analysis and Reliability of Scales
Overall, the statistical approach is sound. The authors report descriptive differences (means, SDs) and effect sizes (Cohen’s d) for each scale comparing the 1990 norms to the current sample. It appears many scales shifted by 1 SD or more (some up to +1.8 SD, i.e. 18 T-score points) in the student sample. I agree with the authors that those are very large differences, leaving little doubt that the old norms misestimate contemporary students’ standing. Still, I have a few suggestions to improve the statistical presentation and address potential weaknesses.

While effect sizes are provided, which is excellent, the manuscript does not explicitly report whether differences are statistically significant. Given the large effect sizes (d values) and the reasonably large sample, most score differences are almost certainly statistically significant. However, the authors stop short of formally testing this—for example, by reporting one-sample t-tests or confidence intervals comparing sample means to the normative mean of T = 50. Including such inferential statistics would have provided clearer support for their claim that the existing norms systematically misclassify typical respondents as pathological.

With 22 main scales and numerous subscales, evaluating deviations from the normative mean inherently involves multiple comparisons, raising concerns about inflated Type I error rates. While the authors focus on effect sizes (i.e., Cohen’s d) rather than formal hypothesis testing, this approach does not fully address the risk of spurious findings across many variables. Effect sizes are informative but do not account for sampling variability or the probability of observing apparent differences by chance when many variables are examined simultaneously. The manuscript would benefit from explicitly acknowledging this limitation, particularly by clarifying whether any scales showed minimal or no deviation from the normative mean (e.g., d ≈ 0). Such transparency would help readers assess whether norm drift is truly widespread or concentrated in specific domains. Including confidence intervals for the reported d values, or highlighting which scales were least affected, would further improve the interpretability and robustness of the findings. To enhance clarity and transparency, the authors should consider including a figure that summarizes the mean deviations from the normative T-score of 50 across all scales—ideally using effect sizes (e.g., Cohen’s d) with 95% confidence intervals. Such a figure would allow readers to visually assess the magnitude and variability of deviations across domains and help identify which scales showed large, small, or negligible changes. This would also mitigate concerns about multiple comparisons by making the pattern of results more interpretable and less reliant on text-based summaries. A well-constructed visual could strengthen the manuscript’s central claim and provide a more balanced view of which aspects of the PAI have or have not shifted since the original norming. In this context, the Discussion notes that effect sizes ranged from –1.0 to +1.8 SD, suggesting that some traits may actually have declined relative to the original norms, possibly on scales like Dominance or Warmth?. It would be valuable if the authors identify any such counter-trends explicitly, as they may reveal important nuances, for instance, that today’s students might report less antisocial or aggressive behavior than in 1990. Clarifying what has remained stable or declined would contribute to a more balanced and comprehensive account of the observed normative shifts.

A significant portion of the manuscript’s argument rests on the claim that some PAI scales demonstrate inadequate reliability (as measured by Cronbach’s α) for high-stakes decision-making. The data presented in Table 1 (and Tables 2–3 for subscales and indexes) support this concern, but the text would benefit from more elaboration. For example, which specific scales had notably low α values? Were these primarily shorter scales or supplementary indexes? If several core clinical scales yielded α values in the 0.60–0.70 range, that does raise concern about measurement error, particularly when scores are used to inform consequential outcomes such as child custody decisions. That said, Cronbach’s α can be deflated in homogeneous samples due to restricted variance, as may be the case with a relatively uniform group of undergraduate psychology students. The authors should acknowledge this limitation and avoid overgeneralizing these estimates to more diverse populations. Beyond this, it’s worth noting that Cronbach’s α has well-documented psychometric limitations. It assumes tau-equivalence, is sensitive to item number and inter-item variance, and may underestimate reliability for multidimensional or heterogenous constructs. Contemporary standards increasingly recommend alternatives such as McDonald’s ω or hierarchical models that provide more accurate reliability estimates. While reporting α remains common, relying on it exclusively, especially for complex, multidimensional assessments like the PAI, is problematic. Nonetheless, the broader point stands, in high-stakes settings, strong reliability is essential to minimize classification error. Professional guidelines, such as the Standards for Educational and Psychological Testing, suggest that reliability coefficients of ≥0.90 are typically expected when tests are used for individual-level decisions. If, as appears likely, many PAI scales fall short of this benchmark, the authors' critique gains further weight. Including this normative context would strengthen their argument and reinforce the ethical implications of using outdated norms alongside modestly reliable scores.

One intriguing finding is that the student sample scored notably high on the Negative Impression Management (NIM) validity scale, while low on Positive Impression (PIM). This suggests that, on average, these students tended to endorse more negative or pathological descriptors about themselves and did not attempt to present themselves favorably. The authors interpret this as part of the norm shift, that is that today’s young adults report more distress and less denial of problems than the 1990 norm group. Another possibility is that response style differences are at play: newer generations might be more open about mental health struggles (reducing PIM, the “fake-good” tendency) and possibly even over-endorse negative feelings (inflating NIM) because of greater awareness of psychological concepts or a culture that encourages sharing one’s troubles. The manuscript should discuss this. If a substantial subset of students had invalidly high NIM scores by traditional standards, the authors should report how they handled those cases. Did they exclude any participants for validity concerns? They mention only “201” students in one analysis, implying a few of the 207 were dropped, perhaps due to invalid profiles. It’s important to ensure that the high base rates of elevated clinical scales aren’t simply an artifact of invalid response sets. Emerging evidence suggests that contemporary college students, particularly those from younger generations, are both more distressed and more open about mental health than previous cohorts. For example, large-scale and longitudinal studies of Norwegian students found a significant increase in self-reported psychological distress (Sivertsen et al., 2023; Knapstad et al., 2021). In North America, surveys have shown that members of Generation Z are significantly more likely to report anxiety or depression compared to older adults and are also more open in discussing psychological symptoms, reflecting declining stigma and increasing mental health literacy (Twenge et al., 2019; American Psychological Association, 2018). These generational shifts may partly explain elevated NIM scores and reduced PIM scores, suggesting not malingering or denial, but greater willingness to disclose distress. This cultural shift reinforces the need to reconsider the interpretation and thresholds of validity scales in light of changing norms. Given that the average NIM was high but presumably not extreme for most, I suspect this reflects a genuine shift in self-report style or mental health. The authors could cite evidence (if any) of rising rates of self-reported distress in college surveys. At minimum, they should caution that elevated clinical scores might partly reflect a general negative response bias, whether due to mood or exaggeration, which itself underscores the need for updated norms. If normal students now routinely score a bit higher on NIM, the old cutoff for “invalid exaggeration” may also be too low, meaning many non-malingering individuals would appear to be over-reporting. This is a critical nuance, the test’s validity indicators and interpretive conventions might need revision alongside the norms.

Interpretation of “Pathology” and Base Rate of High Scores
The authors conclude that “outdated norms identify psychopathology in nearly everyone.” Indeed, their data show 95% of normal students had at least one elevated scale. This over-identification problem is a central critique, and the manuscript does well to highlight it. However, I encourage the authors to fully develop their framing of this issue in terms of multivariate base rates and normative interpretation, as introduced earlier. Doing so would strengthen the case that the observed profile elevations reflect a systemic shift in norms, rather than just isolated scale fluctuations. It is a well-known psychometric fact that when you have many scales, the probability of any one scale being “abnormal” in a given individual is high by chance alone. For example, even if an examinee’s trait levels were perfectly typical, testing them on the 22 PAI-scales means one or two scores will often fall in the top 15% of the distribution purely due to measurement error or normal variation. The authors cite prior work in cognitive testing (Crawford et al., 2007; Karr et al., 2017, etc.) to this effect. They should ensure this concept is clearly explained in their Discussion. Some of that 95% rate is not because today’s students are all suffering latent psychopathology, but because with 22 opportunities, almost everyone will flag on something. In fact, even in 1991 one would expect a sizable minority of the normative sample to have at least one T ≥ 60. The crucial point is that with today’s elevated means, the false-positive problem is magnified. Not only does chance play a role, but the entire score distribution has shifted upward, so that multiple scales per person may exceed old cutoffs.

To illustrate, the authors found that 73% of students had at least one scale at or above 70T, a threshold traditionally interpreted as a “clinical elevation.” Under the original norms, a T-score of 70 corresponds to approximately the 98th percentile. If we assume independence across the 22 main scales, the probability of a person scoring at or above the 98th percentile on at least one scale by chance is roughly 36% 1−(0.98)221−(0.98)22. The observed rate of 73% is more than double that, indicating a systematic shift beyond what would be expected from random variation alone. An explicit analysis comparing these expected versus observed multivariate base rates would further strengthen the argument that outdated norms are not just statistically obsolete, but are actively misclassifying typical individuals as clinically elevated. Such an analysis would help quantify the degree of over-identification that is occurring due to the norm shift, thereby providing a stronger empirical basis for the claim that the current norms systematically misclassify typical respondents as disordered.

Finally, the authors might consider a subtle conceptual point: When a large portion of the population scores in ranges once deemed “clinical,” it challenges the very definition of pathology on these tests. One could argue that if everyone has something, perhaps those traits aren’t truly “abnormal” anymore. The authors lean into this by saying many who show elevated scores are “in fact perfectly normal and average” in today’s context. I partially agree, and this underscores why test norms must be periodically updated to reflect new reality. However, an alternative interpretation is that the prevalence of mental health issues has genuinely increased in modern cohorts, at least for anxiety, depression, etc. If that is so, then the high scores might be accurately flagging distress that just happens to be widespread, an alarming societal issue. The authors should acknowledge this perspective, even if to refute it. For instance, Are today’s students actually more anxious and depressed? Or has openness to admitting symptoms increased? Or are the old norms artificially low due to past under-reporting? Likely all of the above contribute. The key point for the test user, however, is unchanged; whether due to actual symptom prevalence or reporting style, the old norms no longer provide a valid benchmark. A score of 65T that once meant “above average psychopathology” might now be just the average young person’s self-rating on that trait. The manuscript drives this home well; just ensure to discuss the interpretation that rising scores could reflect real trends in psychopathology (with references to rising anxiety in youth, if available) – this will preempt any critics who argue “but what if nearly everyone really is somewhat impaired nowadays?”.

Conclusions and High-Stakes Implications
The authors issue a strong conclusion that the PAI’s outdated norms “ought not to be used in any high stakes assessment and ought to be withdrawn immediately”, and that continued use “likely amounts to malpractice”. I largely agree with their sense of urgency, but the tone and wording here may need careful calibration. Accusing practitioners of malpractice is a serious charge. If this claim is to remain, it should be backed by clear reference to professional standards. The authors do cite ethical guidelines (e.g., APA, AERA & NCME), which frames the argument not as hyperbole but as established ethical dutyusing obsolete norms violates standard practice, potentially harming clients. Perhaps rephrase “amounts to malpractice” to “is professionally unethical and potentially harmful”, which is a bit less provocative but still firm. The goal is to persuade clinicians and test publishers, not alienate them with perceived accusation. Constructively, the authors might add recommendations for moving forward. For example, urging the test publisher to collect new norms, or advising practitioners to use local norms (like the ones the authors provide for Canadian students) or at least interpret PAI results cautiously alongside other evidence. One practical suggestion is interim use of percentile ranks or base rate tables from contemporary samples. The manuscript already includes Tables 4–5 with base rates in the new sample, which is great. By providing those, the authors equip practitioners to say “a T=60 on X scale is actually ~50th percentile among students, not necessarily pathological.” Emphasize this contribution – it adds value beyond just criticism.

Finally, the authors may consider the broader impact on PAI’s validity. If norms are obsolete, does it also call into question the interpretive meaning of certain scales? For instance, is a “moderate elevation” on Borderline Features in a college student actually indicative of borderline personality features, or could it reflect transient stress common to that age? The PAI was originally validated by linking certain T-score ranges to likely clinical implications. Those interpretations might need revising. The authors’ data hint that the PAI’s cut-offs now misclassify many normal individuals as having significant personality pathology. This could have serious consequences in forensic settings , as the authors note, people could unjustly lose opportunities or custody because a test says they have “clinical” levels on some scale. I encourage the authors to drive home this point with concrete examples: e.g., “Using the old norms, a police job applicant scoring 65T on the Anxiety scale would be flagged as unusually anxious, whereas in reality that might be an average level for today’s young adults.” Such scenarios make the implications tangible.

Additional comments

One of the strengths of the manuscript is its focus on an empirical, test-theory issue (i.e., norms and reliability). To further strengthen the theoretical backdrop, the authors should consider connecting their findings to modern models of personality and psychopathology. Over the last few decades, there has been a shift toward viewing psychopathology dimensionally and in continuity with normal personality– an idea that resonates with what the authors found (i.e., that normal individuals often have ostensibly “pathological” traits). Classic and contemporary works by Cloninger and colleagues, for example, can provide a rich context.

Cloninger’s psychobiological model posits that normal vs. abnormal is largely a matter of quantitative difference in personality organization, rather than a strict dichotomy. In a seminal paper, Cloninger and colleagues (1997) argued that “the risk for development of nearly all psychopathology may be precisely predicted if personality is quantified as a self-organizing multidimensional complex”. Essentially, abnormal personality development underlies most mental disorders. This perspective implies that extreme trait scores, like very high anxiety or depression tendencies, are part of a continuous distribution in the population, not categorically separate from “normality.” The authors’ finding that almost everyone has some elevated trait is consistent with this, it may simply reflect that each person has a unique configuration of personality traits, some of which reach extreme levels without constituting clinical disorder. Incorporating this viewpoint in the Discussion would strengthen the manuscript’s theoretical grounding. It shows that the authors’ conclusions are not only about test norms but tie into a larger trend in psychopathology research, moving away from rigid normal/pathological cut-offs toward individualized trait profiles.

The manuscript specifically flags the PAI Schizophrenia scale as one that showed large elevations in students (mean ~60T). We should be cautious in interpreting what an elevated “schizophrenia” scale means in a non-clinical sample– likely it reflects attenuated psychotic-like experiences or eccentricity rather than true psychotic disorder. Recent genetic and neurobiological research has confirmed that broad constructs like schizophrenia are highly heterogeneous. Arnedo and colleagues (2015), for instance, demonstrated that “schizophrenia is a group of heritable disorders caused by a moderate number of separate genotypic networks associated with several distinct clinical syndromes”. In other words, what we call “schizophrenia” is actually an umbrella for multiple subtypes and trait constellations. A single PAI scale cannot capture this nuance, a moderately high score might indicate a variety of benign or transient trait expressions (e.g. imaginative thinking, social detachment) rather than a categorical illness. The authors might use this evidence to caution against over-interpreting any single elevated clinical scale. Modern psychopathology models (including the emerging Hierarchical Taxonomy of Psychopathology, HiTOP) emphasize that individuals often have mixes of traits that don’t neatly map onto traditional diagnoses. Thus, seeing many normals with one or two elevated PAI scales is not surprising. It aligns with the idea that psychopathology traits are continuously distributed and not all-or-none. Citing these or similar work underscores that the PAI’s structure, based on 1990s diagnostic categories, may itself be outdated, not just its norms.

The authors highlight how virtually any PAI profile nowadays shows some “abnormal” spike. This could be reframed as “nearly everyone has some risk factor or vulnerability”. Contemporary research by Cloninger and others supports this. In a recent study (Zwir et al., 2023) identified distinct brain connectivity and personality trait configurations that predispose people to psychosis-spectrum disorders. Interestingly, they found that clinically stable patients with different diagnoses (schizophrenia vs. bipolar) were highly similar in underlying personality and connectivity profiles, differing only in how stress triggered their symptoms. This lends evidence to a diathesis-stress model: many individuals share latent trait vulnerabilities (diatheses) that might never manifest as disorder unless activated by stress. From this viewpoint, it’s expected that a “normal” population will show a lot of subclinical elevations, these could be viewed as latent vulnerabilities or personality extremes rather than immediate pathology. The authors might integrate this idea to validate their concern: a high PAI score does not equal current disorder. It might signal a vulnerability (e.g. high trait anxiety) which, in a supportive environment, does not cause dysfunction. Using updated models like this, the authors can argue that test norms must be interpreted in light of what we now know: psychopathology lies on continua, and context and combinations of traits matter. This is something a simple norm-referenced T-score alone cannot capture.

In sum, relating the findings to such literature will enrich the discussion. It shows that the authors are aware of broader developments in personality science, namely that the boundary between normal personality variation and psychopathology is more porous than old categorical tests assume. This bolsters their call for caution: a test that over-pathologizes is especially problematic today, when theory tells us that many so-called “pathological” traits are common in the general population. The authors should consider citing research that has proposed integrative, biopsychosocial models. This will help convince readers, and the field, that the issues with the PAI are not just a quirk of one test, but part of a larger need to modernize assessment to align with current science.

References
American Psychological Association. (2018). Stress in America: Generation Z. https://www.apa.org/news/press/releases/stress/2018/stress-gen-z.pdf

Arnedo, J., Svrakic, D. M., del Val, C., Romero-Zaliz, R., Hernández-Cuervo, H., … Zwir, I. (2015). Uncovering the Hidden Risk Architecture of the Schizophrenias: Confirmation in Three Independent Genome-Wide Association Studies. American Journal of Psychiatry, 172(2), 139–153. https://doi.org/10.1176/appi.ajp.2014.14040435

Bayram, N., Bilgel, N. (2008). The prevalence and socio-demographic correlations of depression, anxiety and stress among a group of university students. Social Psychiatry and Epidemiology, 43, 667–672. https://doi.org/10.1007/s00127-008-0345-x

Cloninger, C. R., Svrakic, N. M., & Svrakic, D. M. (1997). Role of personality self-organization in development of mental order and disorder. Development and psychopathology, 9(4), 881–906. https://doi.org/10.1017/s095457949700148x

Dong, Y., & Dumas, D. (2020). Are personality measures valid for different populations? A systematic review of measurement invariance across cultures, gender, and age. Personality and Individual Differences, 160, 109956. https://doi.org/10.1016/j.paid.2020.109956

Ibrahim, A. K., Kelly, S. J., Adams, C. E., & Glazebrook, C. (2013). A systematic review of studies of depression prevalence in university students. Journal of Psychiatric Research, 47(3), 391–400. https://doi.org/10.1016/j.jpsychires.2012.11.015

Knapstad, M., Sivertsen, B., Knudsen, A. K., Smith, O. R. F., Aarø, L. E., Lønning, K. J., & Skogen, J. C. (2021). Trends in self‑reported psychological distress among college and university students from 2010 to 2018. Psychological Medicine, 51(3), 470–478. https://doi.org/10.1017/S0033291719003350

Nails, J. G., Maffly-Kipp, J., DeShong, H. L., Lowmaster, S. E., & Kurtz, J. E. (2023). A crisis in college student mental health? Self-ratings of psychopathology before and after the COVID-19 pandemic. Psychological Assessment, 35(11), 101031018. https://doi.org/10.1037/pas0001241

Schmitt, D. P., Allik, J., McCrae, R. R., & Benet-Martínez, V. (2007). The geographic distribution of Big Five personality traits: Patterns and profiles of human self‑description across 56 nations. Journal of Cross-Cultural Psychology, 38(2), 173–212. https://doi.org/10.1177/0022022106297299

Sivertsen, B., Skrindo Knudsen, A-K., Kirkøen, B., et al. (2023). Prevalence of mental disorders among Norwegian college and university students: a population-based cross-sectional analysis. The Lancet Regional Health, 34, 100732. https://doi.org/10.1016/j.lanepe.2023.100732

Twenge, J. M., Cooper, A. B., Joiner, T. E., Duffy, M. E., & Binau, S. G. (2019). Age, period, and cohort trends in mood disorder indicators and suicide-related outcomes in a nationally representative dataset, 2005–2017. Journal of Abnormal Psychology, 128(3), 185–199. https://doi.org/10.1037/abn0000410

Zwir, I., Arnedo, J., Mesa, A. et al. Temperament & Character account for brain functional connectivity at rest: A diathesis-stress model of functional dysregulation in psychosis. Molecular Psychiatry, 28, 2238–2253 (2023). https://doi.org/10.1038/s41380-023-02039-6

·

Basic reporting

In general, this paper is clear, well-argued with evidence to support the arguments, and shows good use of the literature. However, I do see some redundancy in the test; in particular, lines 74-94 and lines 95-118, are highly overlapping and could be combined, with an increase in comprehensibility. It is strongly worded, which is typical for this group, and that is a matter of personal preference.

Some greater specificity of references would be useful. For example, line 520; Lilienfeld et al. is a book, so there should be a specific page reference for the citation. There are 3 papers by Karr et al. cited, and at least one could be deleted without affecting the argument.

Experimental design

Overall, the experimental design appears appropriate. One issue, however, is that NIM appears not to have been used as a validity scale. Nails et al removed all cases with a NIM score of 92 or greater, as evidence of pronounced reporting bias, and for comparison, such a cutoff may be helpful in this study. It appears from Figure 1 that this would eliminate about 8 cases, and would increase comparison to other studies.

Validity of the findings

In my opinion, these findings are strong on one aspect, but need elaboration in another.
First, the mean changes (or perhaps better stated, variability between samples and cohorts) is strongly supported by this paper and findings. This needs to be considered on a situational level, and comparison to one set of benchmarks (the 1991 college sample, for example) may not be appropriate.
The second point, which deserves more elaboration, is how the statistical definition of abnormality (scores at a certain distance from the norm) fits, or does not fit, with the pathological definition (i.e., causing significant distress to the individual). If rates of diagnosed anxiety disorders or depression are increasing among university students, for example (Nails et al., 2023), then an increased number of students should show scores beyond a threshold, or the test is not identifying an important characteristic. I do find this aspect is not well-described in the current paper, and considering this would greatly strengthen the paper and conclusions. These articles might inform consideration on this topic.
Auerbach, R. P., Mortier, P., Bruffaerts, R., Alonso, J., Benjet, C., Cuijpers, P., ... & Kessler, R. C. (2018). WHO world mental health surveys international college student project: Prevalence and distribution of mental disorders. Journal of abnormal psychology, 127(7), 623.
Buizza, C., Bazzoli, L., & Ghilardi, A. (2022). Changes in college students mental health and lifestyle during the COVID-19 pandemic: a systematic review of longitudinal studies. Adolescent research review, 7(4), 537-550.

Additional comments

Overall, I think this paper is important, but will benefit from greater attention to what changes in scores would be expected, as these tools do not exist in a vacuum. I do appreciate the attention paid to the psychometric aspects of this tool, which is often not considered sufficiently, and in particular, examples of lower reliabilities, and how this might affect interpretation.
There are some redundancies in language, which could be removed; e.g., line 90, 'outdated, obsolete', or line 326 'normal, average'.
I appreciate the identification that a large minority of the general population would be identified as minimizing/faking good, based on the PIM scale. This is a problem when the PAI is broadly used, and not restricted to a pathological population, and benefits from being considered.

---

## Round 0.2 · Minor Revisions

· Academic Editor

Minor Revisions

You have made the paper much more interesting and important in your first round of revisions. All that remains now is to revise the language using Reviewer1's excellent constructive suggestions as well as Reviewer 2's, and adding some points and clarifications as suggested by both reviewers.

Reviewer 1 ·

Basic reporting

Please see my comments at the end.

Experimental design

Please see my comments at the end.

Validity of the findings

Please see my comments at the end.

Additional comments

Overall, the revised manuscript is far more robust and clear in conveying its central message that the PAI’s norms are outdated and can misidentify healthy individuals as pathological. The authors have been highly responsive to critiques across methodological, statistical, and interpretive domains. The improvements, from acknowledging sample limits to providing vivid figures and deeper discussion, have strengthened the paper. The review comments appear to have been addressed effectively and in good faith. I commend the authors for their comprehensive revisions and believe the manuscript is much improved as a result.

The remaining critiques are few and mostly stylistic or interpretive. My main suggestion is that the authors consider softening some of the rhetorical language, particularly where the manuscript uses very authoritative or absolute phrasing about ethics and practitioner behavior. For instance, the abstract currently concludes with:

“We conclude that the PAI US norms ought not to be used in any high stakes assessment and ought to be withdrawn immediately to avoid harm to the examinees and to society. The continued use of the PAI US outdated, obsolete, and invalid norms in high stakes assessments is non-scientific and likely amounts to malpractice.”

While this reflects the authors’ strong views, the tone may come across as accusatory or overly prescriptive. The same idea could be communicated in a more collaborative way, such as:

“We conclude that the PAI US norms are no longer appropriate for high-stakes assessments and should be updated as a matter of urgency to prevent misclassification and potential harm to examinees. Continued use of outdated norms carries ethical and scientific risks and may lead to serious consequences in practice.”

This alternative maintains the scientific and ethical concerns but frames them in a way that invites dialogue rather than conflict. Moderating the tone in key places throughout the manuscript, particularly in the abstract, discussion, and recommendations, would make the paper more persuasive and accessible to a wider audience, including practitioners and policymakers.

I strongly recommend these adjustments as they will enhance the manuscript’s impact and readability. The paper will serve as a highly valuable, well-supported contribution to the ongoing conversation about psychological test norms and their shelf life, both for researchers and practicing clinicians.

·

Basic reporting

Overall, the paper is clear, unambiguous, and provides sufficient background and reference to the literature. The structure is clear, figures and tables are relevant and appropriate.

Experimental design

This is a descriptive study, and is clearly explained, examining how test scores differ between a Canadian sample and American comparison groups. Procedureis clear.

Validity of the findings

Underlying data are presented and characterized (although not provided) and are Conceptually appropriate. Conclusions are based on results of applying standard interpretive guidelines, and thus follow the data.

Additional comments

The authors have incorporated recommendations from previous review, with some increased discussion about potential causes for changes, and some additional statistical reporting (although this has shown little change in results. Tables 4 and 5were particularly useful, as are figures 1, 3 and 4. There are a few minor typos or wording changes that I will suggest.

At the risk of expanding this paper, as a clinician I would note that the common alternative to the PAI, the MMPI-3, also suffers from lack of Canadian norms and validation in Canada, although the norms are more recent.

Copy editing
Line 91. Add “original” before “US normative college sample”
Line 140. “Despite all of the above” can be omitted
Line 155. “US norms were a poor characterization of” suggested as alternate wording
line 213: suggest: “2-hour long” as clearer wording
Line 460: note PAI Plus continues to use original norms.
Line 585: necessity
Line 856: Russell

---

## Round 0.3 · accepted · Accept

· Academic Editor

Accept

I hope this paper will be seen as a call to action, for the PAI and for many other psychological tests. I would have preferred that you soften your tone - because in my experience this level of anger and accusation (although justified) often does not produce constructive results. Let us wait and see if the paper makes the ripples we are hoping for.